# LS-IQ: Implicit Reward Regularization for Inverse Reinforcement Learning

**Firas Al-Hafez[1], Davide Tateo[1], Oleg Arenz[1], Guoping Zhao[2], Jan Peters[1,3]**
[1] Intelligent Autonomous Systems, [2] Locomotion Laboratory
[3] German Research Center for AI (DFKI), Centre for Cognitive Science, Hessian.AI
TU Darmstadt, Germany
{name.surname}@tu-darmstadt.de

## Abstract

Recent methods for imitation learning directly learn a $Q$-function using an implicit reward formulation rather than an explicit reward function. However, these methods generally require implicit reward regularization to improve stability and often mistreat absorbing states. Previous works show that a squared norm regularization on the implicit reward function is effective, but do not provide a theoretical analysis of the resulting properties of the algorithms. In this work, we show that using this regularizer under a mixture distribution of the policy and the expert provides a particularly illuminating perspective: the original objective can be understood as squared Bellman error minimization, and the corresponding optimization problem minimizes a bounded $\chi^2$-Divergence between the expert and the mixture distribution. This perspective allows us to address instabilities and properly treat absorbing states. We show that our method, Least Squares Inverse Q-Learning (LS-IQ), outperforms state-of-the-art algorithms, particularly in environments with absorbing states. Finally, we propose to use an inverse dynamics model to learn from observations only. Using this approach, we retain performance in settings where no expert actions are available.[1]

## 1 Introduction

Inverse Reinforcement Learning (IRL) techniques have been developed to robustly extract behaviors from expert demonstration and solve the problems of classical Imitation Learning (IL) methods (Ng et al., 1999; Ziebart et al., 2008). Among the recent methods for IRL, the Adversarial Imitation Learning (AIL) approach (Ho & Ermon, 2016; Fu et al., 2018; Peng et al., 2021), which casts the optimization over rewards and policies into an adversarial setting, have been proven particularly successful. These methods, inspired by Generative Adversarial Networks (GANs) (Goodfellow et al., 2014), alternate between learning a discriminator, and improving the agent's policy w.r.t. a reward function, computed based on the discriminator's output. These *explicit reward* methods require many interactions with the environment as they learn both a reward and a value function. Recently, *implicit reward* methods (Kostrikov et al., 2020; Arenz & Neumann, 2020; Garg et al., 2021) have been proposed. These methods directly learn the $Q$-function, significantly accelerating the policy optimization. Among the *implicit reward* approaches, the Inverse soft Q-Learning (IQ-Learn) is the current state-of-the-art. This method modifies the distribution matching objective by including reward regularization on the expert distribution, which results in a minimization of the $\chi^2$-divergence between the policy and the expert distribution. However, whereas their derivations only consider regularization on the expert distribution, their practical implementations on continuous control tasks have shown that regularizing the reward on both the expert and policy distribution achieves significantly better performance.

The contribution of this paper is twofold: First, when using this regularizer, we show that the resulting objective minimizes the $\chi^2$ divergence between the expert and a mixture distribution between the expert and the policy. We then investigate the effects of regularizing w.r.t. the mixture distribution on the theoretical properties of IQ-Learn. We show that this divergence is bounded, which translates

---

[1]The code is available at https://github.com/robfiras/ls-iq

to bounds on the reward and $Q$-function, significantly improving learning stability. Indeed, the resulting objective corresponds to least-squares Bellman error minimization and is closely related to Soft Q-Imitation Learning (SQIL) (Reddy et al., 2020). Second, we formulate Least Squares Inverse Q-Learning (LS-IQ), a novel IRL algorithm. By following the theoretical insight coming from the analysis of the $\chi^2$ regularizer, we tackle many sources of instabilities of the IQ-Learn approach: the arbitrariness of the $Q$-function scales, exploding $Q$-functions targets, and reward bias Kostrikov et al. (2019), i.e., assuming that absorbing states provide the null reward. We derive the LS-IQ algorithm by exploiting structural properties of the $Q$-function and heuristics based on expert optimality. This results in increased performance on many tasks and, in general, more stable learning and less variance in the $Q$-function estimation. Finally, we extend the implicit reward methods to the IL from observations setting by training an Inverse-Dynamics Model (IDM) to predict the expert actions, which are no longer assumed to be available. Even in this challenging setting, our approach retains performance similar to the one where expert actions are known.

**Related Work.** The vast majority of IRL and IL methods build upon the Maximum Entropy (MaxEnt) IRL framework (Ziebart, 2010). In particular, Ho & Ermon (2016) introduce Generative Adversarial Imitation Learning (GAIL), which applies GANs to the IL problem. While the original method minimizes the Jensen-Shannon divergence to the expert distribution, the approach is extended to general $f$-divergences (Ghasemipour et al., 2019), building on the work of Nowozin et al. (2016). Among the $f$-divergences, the Pearson $\chi^2$ divergence improves the training stability for GANs (Mao et al., 2017) and for AIL (Peng et al., 2021). Kostrikov et al. (2019) introduce a replay buffer for off-policy updates of the policy and discriminator. The authors also point out the problem of reward bias, which is common in many imitation learning methods. Indeed, AIL methods implicitly assign a null reward to these states, leading to survival or termination biases, depending on the chosen divergence. Kostrikov et al. (2020) improve the previous work introducing recent advances from offline policy evaluation (Nachum et al., 2019). Their method, ValueDice, uses an inverse Bellman operator that expresses the reward function in terms of its $Q$-function, to minimize the reverse Kullback-Leibler Divergence (KL) to the expert distribution. Arenz & Neumann (2020) derive a non-adversarial formulation based on trust-region updates on the policy. Their method, O-NAIL, uses a standard Soft-Actor Critic (SAC) (Haarnoja et al., 2018) update for policy improvement. O-NAIL can be understood as an instance of the more general IQ-Learn algorithm (Garg et al., 2021), which can optimize different divergences depending on an implicit reward regularizer. Garg et al. (2021) also show that their algorithm achieves better performance using the $\chi^2$ divergence instead of the reverse KL. Reddy et al. (2020) propose a method that uses SAC and assigns fixed binary rewards to the expert and the policy. Swamy et al. (2021) provide a unifying perspective on many of the methods mentioned above, explicitly showing that GAIL, ValueDice, MaxEnt-IRL, and SQIL can be viewed as moment matching algorithms. Lastly, Sikchi et al. (2023) propose a ranking loss for AIL, which trains a reward function using a least-squares objective with ranked targets.

## 2 Preliminaries

**Notation.** A Markov Decision Process (MDP) is a tuple $(\mathcal{S}, \mathcal{A}, P, r, \gamma, \mu_0)$, where $\mathcal{S}$ is the state space, $\mathcal{A}$ is the action space, $P : \mathcal{S} \times \mathcal{A} \times \mathcal{S} \to \mathbb{R}^+$ is the transition kernel, $r : \mathcal{S} \times \mathcal{A} \to \mathbb{R}$ is the reward function, $\gamma$ is the discount factor, and $\mu_0 : \mathcal{S} \to \mathbb{R}^+$ is the initial state distribution. At each step, the agent observes a state $s \in \mathcal{S}$ from the environment, samples an action $a \in \mathcal{A}$ using the policy $\pi : \mathcal{S} \times \mathcal{A} \to \mathbb{R}^+$, and transitions with probability $P(s'|s, a)$ into the next state $s' \in \mathcal{S}$, where it receives the reward $r(s, a)$. We define an occupancy measure $\rho_\pi(s, a) = \pi(a|s) \sum_{t=0}^{\infty} \gamma^t \mu_t^\pi(s)$, where $\mu_t^\pi(s') = \int_{s,a} \mu_t^\pi(s) \pi(a|s) P(s'|s, a) da \, ds$ is the state distribution for $t > 0$, with $\mu_0^\pi(s) = \mu_0(s)$. The occupancy measure allows us to denote the expected reward under policy $\pi$ as $\mathbb{E}_{\rho_\pi}[r(s, a)] \triangleq \mathbb{E}[\sum_{t=0}^{\infty} \gamma^t r(s_t, a_t)]$, where $s_0 \sim \mu_0$, $a_t \sim \pi(.|s_t)$ and $s_{t+1} \sim P(.|s_t, a_t)$ for $t > 0$. Furthermore, $\mathbb{R}^{\mathcal{S} \times \mathcal{A}} = \{x : \mathcal{S} \times \mathcal{A} \to \mathbb{R}\}$ denotes the set of functions in the state-action space and $\overline{\mathbb{R}}$ denotes the extended real numbers $\mathbb{R} \cup \{+\infty\}$. We refer to the soft value functions as $\tilde{V}(s)$ and $\tilde{Q}(s, a)$, while we use $V(s)$ and $Q(s, a)$ to denote the value functions without entropy bonus.

**Inverse Reinforcement Learning as an Occupancy Matching Problem.** Given a set of demonstrations consisting of states and actions sampled from an expert policy $\pi_E$, IRL aims at finding a reward function $r(s, a)$ from a family of reward functions $\mathcal{R} = \mathbb{R}^{\mathcal{S} \times \mathcal{A}}$ assigning high reward to

samples from the expert policy $\pi_E$ and low reward to other policies. We consider the framework presented in Ho & Ermon (2016), which derives the maximum entropy IRL objective with an additional convex reward regularizer $\psi_\rho : \mathbb{R}^{\mathcal{S} \times \mathcal{A}} \to \overline{\mathbb{R}}$ from an occupancy matching problem

$$\max_{r \in \mathcal{R}} \min_{\pi \in \Pi} L_\rho(r, \pi) = \max_{r \in \mathcal{R}} \left( \min_{\pi \in \Pi} -\beta H_\rho(\pi) - \mathbb{E}_{\rho_\pi}[r(s,a)] \right) + \mathbb{E}_{\rho_{\pi_E}}[r(s,a)] - \psi_\rho(r), \qquad (1)$$

with the space of policies $\Pi = \mathbb{R}^{\mathcal{S} \times \mathcal{A}}$, the discounted cumulative entropy bonus $H_\rho(\pi) = \mathbb{E}_{\rho_\pi}[-\log(\pi(a|s))]$, and a constant $\beta$ controlling the entropy bonus. Note that the inner optimization is a maximum entropy Reinforcement Learning (RL) objective (Ziebart, 2010), for which the optimal policy is given by

$$\pi^*(a|s) = \frac{1}{Z_s} \exp(\tilde{Q}(s,a)), \qquad (2)$$

where $Z_s = \int_{\hat{a}} \exp \tilde{Q}(s, \hat{a}) \, d\hat{a}$ is the partition function and $\tilde{Q}(s,a)$ is the soft action-value function, which is given for a certain reward function by the soft Bellman operator $(\tilde{\mathcal{B}}^\pi \tilde{Q})(s,a) = r(s,a) + \gamma \mathbb{E}_{s' \sim P(.|s,a)} \tilde{V}^\pi(s')$, where $\tilde{V}^\pi(s') = \mathbb{E}_{a \sim \pi(.|s)}[\tilde{Q}(s,a) - \log \pi(a|s)]$.

Garg et al. (2021) transform Equation 1 from reward-policy space to $\tilde{Q}$-policy space using the *inverse* soft Bellman operator $(\tilde{\mathcal{T}}^\pi \tilde{Q})(s,a) = \tilde{Q}(s,a) - \gamma \mathbb{E}_{s' \sim P(.|s,a)} \tilde{V}^\pi(s')$ to get a one-to-one correspondence between the reward and the $\tilde{Q}$-function. This operator allows to change the objective function $L_\rho$ from reward-policy to $Q$-policy space, from now on denoted as $\mathcal{J}_\rho$

$$\max_{r \in \mathcal{R}} \min_{\pi \in \Pi} L_\rho(r, \pi) = \max_{\tilde{Q} \in \tilde{\Omega}} \min_{\pi \in \Pi} \mathcal{J}_\rho(\tilde{Q}, \pi), \qquad (3)$$

where $\tilde{\Omega} = \mathbb{R}^{\mathcal{S} \times \mathcal{A}}$ is the space of $\tilde{Q}$-functions. Furthermore, they use Equation 2 to extract the optimal policy $\pi_{\tilde{Q}}$ given a $\tilde{Q}$-function to drop the inner optimization loop in Equation 1 such that

$$\max_{\tilde{Q} \in \tilde{\Omega}} \mathcal{J}_\rho(\tilde{Q}, \pi_{\tilde{Q}}) = \max_{\tilde{Q} \in \tilde{\Omega}} \mathbb{E}_{\rho_{\pi_E}} \left[ \tilde{Q}(s,a) - \gamma \mathbb{E}_{s' \sim P(.|s,a)}[\tilde{V}^\pi(s')] \right] - \beta H_\rho(\pi_{\tilde{Q}}) \qquad (4)$$
$$- \mathbb{E}_{\rho_\pi} \left[ \tilde{Q}(s,a) - \gamma \mathbb{E}_{s' \sim P(.|s,a)}[\tilde{V}^\pi(s')] \right] - \psi_\rho(\tilde{\mathcal{T}}^\pi \tilde{Q}).$$

**Practical Reward Regularization.** Garg et al. (2021) derive a regularizer enforcing an $L_2$ norm-penalty on the reward on state-action pairs from the expert, such that $\psi_{\pi_E}(r) = c \, \mathbb{E}_{\rho_{\pi_E}} \left[ r(s,a)^2 \right]$ with $c$ being a regularizer constant. However, in continuous action spaces, this regularizer causes instabilities. In practice, Garg et al. (2021) address this instabilities by using the regularizer to the mixture

$$\psi_\rho(r) = \alpha \, c \, \mathbb{E}_{\rho_{\pi_E}} \left[ r(s,a)^2 \right] + (1 - \alpha) \, c \, \mathbb{E}_{\rho_\pi} \left[ r(s,a)^2 \right], \qquad (5)$$

where $\alpha$ is typically set to $0.5$. It is important to note that this change of regularizer does not allow the direct extraction of the policy from Equation 1 anymore. Indeed, the regularizer in Equation 5 also depends on the policy. Prior work did not address this issue. In the following sections, we will provide an in-depth analysis of this regularizer, allowing us to address the aforementioned issues and derive the correct policy update. Before we introduce our method, we use Proposition A.1 in Appendix A to change the objectives $L_\rho$ and $\mathcal{J}_\rho$ from expectations under occupancy measures to expectations under state-action distributions $d_{\pi_E}$ and $d_\pi$, from now on denoted as $L$ and $\mathcal{J}$, respectively.

## 3 Least Squares Inverse Q-Learning

In this section, we introduce our proposed imitation learning algorithm, which is based on the occupancy matching problem presented in Equation 1 using the regularizer defined in Equation 5. We start by giving an interpretation of the resulting objective as a $\chi^2$ divergence between the expert distribution and a mixture distribution of the expert and the policy. We then show that the regularizer allows us to cast the original objective into a Bellman error minimization problem with fixed binary rewards for the expert and the policy. An RL problem with fixed rewards is a unique setting, which we can utilize to bound the $Q$-function target, provide fixed targets for the $Q$-function on expert states instead of doing bootstrapping, and adequately treat absorbing states. However, these techniques need to be applied on hard $Q$-functions. Therefore, we switch from soft action-value functions $\tilde{Q}$ to hard $Q$-functions, by introducing an additional entropy critic. We also present a regularization critic allowing us to recover the correct policy update corresponding to the regularizer in Equation 5. Finally, we propose to use an IDM to solve the imitation learning from observations problem.

## 3.1 Interpretation as a Statistical Divergence

Ho & Ermon (2016) showed that their regularizer results in a Jensen-Shannon divergence minimization between the expert's and the policy's state-action distribution. Similarily, Garg et al. (2021) showed that their regularizer $\psi_{\pi_E}(r)$ results in a minimization of the $\chi^2$ divergence. However, the regularizer presented in Equation 5 is not investigated yet. We show that this regularizer minimizes a $\chi^2$ divergence between the expert's state-action distribution and a mixture distribution between the expert and the policy. Therefore, we start with the objective presented in Equation 1 and note that strong duality $-\max_{r \in \mathcal{R}} \min_{\pi \in \Pi} L = \min_{\pi \in \Pi} \max_{r \in \mathcal{R}} L$ – follows straightforwardly from the minimax theorem (Von Neumann, 1928) as $-H(\pi)$, $-\mathbb{E}_{d_\pi}[r(s,a)]$ and $\psi(r)$ are convex in $\pi$, and $-\mathbb{E}_{d_\pi}[r(s,a)]$, $\mathbb{E}_{d_{\pi_E}}[r(s,a)]$ and $\psi(r)$ are concave in $r$ (Ho & Ermon, 2016). We express the $\chi^2$ divergence between the expert's distribution and the mixture distribution using its variational form,

$$2\chi^2\Big(d_{\pi_E} \,\big\|\, \underbrace{\tfrac{d_{\pi_E}+d_\pi}{2}}_{d_{\mathrm{mix}}}\Big) = \sup_r 2\Big(\mathbb{E}_{d_{\pi_E}}[r(s,a)] - \mathbb{E}_{\tilde{d}}\Big[r(s,a) + \tfrac{r(s,a)^2}{4}\Big]\Big)$$
$$= \sup_r \mathbb{E}_{d_{\pi_E}}[r(s,a)] - \mathbb{E}_{d_\pi}[r(s,a)] - c\alpha\mathbb{E}_{d_{\pi_E}}[r(s,a)^2] - c(1-\alpha)\mathbb{E}_{d_\pi}[r(s,a)^2], \quad (6)$$

with the regularizer constant $c = 1/2$ and $\alpha = 1/2$. Now, if the optimal reward is in $\mathcal{R}$, the original objective from Equation 1 becomes an entropy-regularized $\chi^2$ divergence minimization problem

$$\max_{r \in \mathcal{R}} \min_{\pi \in \Pi} L = \min_{\pi \in \Pi} 2\chi^2\big(d_{\pi_E} \,\big\|\, \tfrac{d_{\pi_E}+d_\pi}{2}\big) - \beta H(\pi). \quad (7)$$

Equation 7 tells us that the regularized IRL objective optimizes the reward to match a divergence while optimizing the policy to minimize the latter. When the divergence to be matched is unbounded, the optimal reward is also unbounded, causing instability during learning. Unlike the $\chi^2$-divergence between the agent's and the expert's distribution, the $\chi^2$-divergence to the mixture distribution is bounded to $[0, 1/c]$ as shown in Proposition A.3, and its optimal reward

$$r^*(s,a) = \frac{1}{c}\frac{d_{\pi_E}(s,a) - d_\pi(s,a)}{d_{\pi_E}(s,a) + d_\pi(s,a)}, \quad (8)$$

is also bounded in the interval $[-1/c, 1/c]$ as shown in Proposition A.2.

## 3.2 A Reinforcement Learning Perspective on Distribution Matching

In the following, we present a novel perspective on Equation 4 allowing us to better understand the effect of the regularizer. Indeed, for the regularizer defined in Equation 5, we can interpret this objective as an entropy-regularized least squares problem, as shown by the following proposition:

**Proposition 3.1** *Let $r_{\tilde{Q}}(s,a) = (\tilde{\mathcal{T}}^\pi \tilde{Q})(s,a)$ be the implicit reward function of a $\tilde{Q}$-function, then for $\psi(r_{\tilde{Q}}) = c\,\mathbb{E}_{\tilde{d}}[r_{\tilde{Q}}(s,a)^2]$ with $\tilde{d}(s,a) = \alpha d_{\pi_E}(s,a) + (1-\alpha)d_\pi(s,a)$, the solution of Equation 4 under state-action distributions equals the solution of an entropy-regularized least squares minimization problem such that $\arg\min_{\tilde{Q} \in \tilde{\Omega}} \mathcal{L}(\tilde{Q}, \pi_{\tilde{Q}}) = \arg\max_{\tilde{Q} \in \tilde{\Omega}} \mathcal{J}(\tilde{Q}, \pi_{\tilde{Q}})$ with*

$$\mathcal{L}(\tilde{Q}, \pi_{\tilde{Q}}) = \alpha\mathbb{E}_{d_{\pi_E}}\Big[\big(r_{\tilde{Q}}(s,a) - r_{\max}\big)^2\Big] + (1-\alpha)\mathbb{E}_{d_{\pi_{\tilde{Q}}}}\Big[\big(r_{\tilde{Q}}(s,a) - r_{\min}\big)^2\Big] + \frac{\beta}{c}H(\pi_{\tilde{Q}}), \quad (9)$$

*where $r_{\max} = \frac{1}{2\alpha c}$ and $r_{\min} = -\frac{1}{2(1-\alpha)c}$.*

The proof is provided in Appendix A.3. The resulting objective in Equation 9 is very similar to the one in the Least Squares Generative Adversarial Networks (LSGANs) (Mao et al., 2017) setting, where $r_{\tilde{Q}}(s,a)$ can be interpreted as the discriminator, $r_{\max}$ can be interpreted as the target for expert samples, and $r_{\min}$ can be interpreted as the target for samples under the policy $\pi$. For $\alpha = 0.5$ and $c = 1$, resulting in $r_{\max} = 1$ and $r_{\min} = -1$, Equation 9 differs from the discriminator's objective in the LSGANs setting only by the entropy term.

Now replacing the implicit reward function with the inverse soft Bellman operator and rearranging the terms yields

$$\mathcal{L}(\tilde{Q}, \pi_{\tilde{Q}}) = \alpha\mathbb{E}_{d_{\pi_E}}\Big[\big(\tilde{Q}(s,a) - (r_{\max} + \gamma\mathbb{E}_{s' \sim P(\cdot|s,a)}[\tilde{V}^\pi(s')])\big)^2\Big] \quad (10)$$

$$+ (1-\alpha)\mathbb{E}_{d_{\pi_{\tilde{Q}}}}\Big[\big(\tilde{Q}(s,a) - (r_{\min} + \gamma\mathbb{E}_{s' \sim P(\cdot|s,a)}[\tilde{V}^\pi(s')])\big)^2\Big] + \frac{\beta}{c}H(\pi_{\tilde{Q}})$$

$$= \alpha\,\delta^2(d_{\pi_E}, r_{\max}) + (1-\alpha)\,\delta^2(d_\pi, r_{\min}) + \frac{\beta}{c}H(\pi_{\tilde{Q}}), \quad (11)$$

where $\delta^2$ is the squared soft Bellman error. We can deduce the following from Equation 11:

$\chi^2$**-regularized IRL under a mixture can be seen as an RL problem** with fixed rewards $r_{\max}$ and $r_{\min}$ for the expert and the policy. This insight allows us to understand the importance of the regularizer constant $c$: it defines the target rewards and, therefore, the scale of the $Q$-function. The resulting objective shows strong relations to the SQIL algorithm, in which also fixed rewards are used. However, SQIL uses $r_{\max} = 1$ and $r_{\min} = 0$, which is infeasible in our setting for $\alpha < 1$. While the entropy term appears to be another difference, we note that it does not affect the critic update, where $\pi_{\tilde{Q}}$ is fixed. As in SQIL, the entropy is maximized by extracting the MaxEnt policy using Equation 2.

**Stabilizing the training in a fixed reward setting is straightforward.** We can have a clean solution to the reward bias problem – c.f., Section 3.4 –, and we can provide fixed $Q$-target for the expert and clipped $Q$-function targets for the policy – c.f., Section 3.5 & 3.7 to improve learning stability significantly. However, we must switch from soft to hard action-value functions by introducing an entropy critic to apply these techniques. Additionally, we show how to recover the correct policy update corresponding to the regularizer in Equation 5 by introducing a regularization critic.

### 3.3 Entropy and Regularization Critic

We express the $\tilde{Q}$-function implicitly using $\tilde{Q}(s, a) = Q(s, a) + \mathcal{H}^\pi(s, a)$ decomposing it into a hard $Q$-function and an *entropy critic*

$$\mathcal{H}^\pi(s, a) = \mathbb{E}_{P,\pi} \left[ \sum_{t'=t}^{\infty} -\gamma^{t'-t+1} \beta \log \pi(a_{t'+1}|s_{t'+1}) \middle| s_t = s, a_t = a \right]. \tag{12}$$

This procedure allows us to stay in the MaxEnt formulation while retaining the ability to operate on the hard $Q$-function. We replace the soft inverse Bellman operator with the hard optimal inverse Bellman operator $(\mathcal{T}Q)(s, a) = Q(s, a) - \gamma \mathbb{E}_{s' \sim P(.|s,a)} V^*(s')$, with the optimal value function $V^*(s) = \max_a Q(s, a)$.

As mentioned before, the regularizer introduced in Equation 5 incorporates yet another term depending on the policy. Indeed, the inner optimization problem in Equation 1—the term in the brackets—is not purely the MaxEnt problem anymore, but includes the term $-k\mathbb{E}_{d_\pi}[r(s, a)^2]$ with $k = c(1-\alpha)$. To incorporate this term into our final implicit action-value function $Q^\dagger(s, a)$, we learn an additional *regularization critic*

$$\mathcal{C}(s, a) = \mathbb{E}_{P,\pi} \left[ \sum_{t'=t}^{\infty} \gamma^{t'-t} r(s_{t'}, a_{t'})^2 \middle| s_t = s, a_t = a \right]. \tag{13}$$

such that $Q^\dagger(s, a) = Q(s, a) + \mathcal{H}^\pi(s, a) + k\,\mathcal{C}(s, a)$. Using $Q^\dagger$, we obtain the exact solution to the inner minimization problem in Equation 2. In practice, we learn a single critic $\mathcal{G}^\pi$ combining $\mathcal{H}^\pi$ and $\mathcal{C}$. We train the latter independently using the following objective

$$\min_{\mathcal{G}^\pi} \delta_{\mathcal{G}}^2 = \min_{\mathcal{G}^\pi} \mathbb{E}_{d_\pi} \left[ (\mathcal{G}^\pi(s, a) - (k\,r_Q(s, a)^2 + \mathbb{E}_{\substack{s' \sim P \\ a' \sim \pi}} [\gamma(-\beta \log \pi(a'|s') + \mathcal{G}^\pi(s', a'))]))^2 \right], \tag{14}$$

which is an entropy-regularized Bellman error minimization problem given the squared implicit reward $r_Q$ scaled by $k$.

### 3.4 Treatment of Absorbing States

Another technical aspect neglected by IQ-Learn is the proper treatment of absorbing states. Garg et al. (2021) treat absorbing states by adding an indicator $\nu$—where $\nu = 1$ if $s'$ is a terminal state—in front of the discounted value function in the inverse Bellman operator

$$(\mathcal{T}_{\text{iq}}^\pi Q)(s, a) = Q(s, a) - (1 - \nu)\gamma \mathbb{E}_{s' \sim P(.|s,a)} V^\pi(s'). \tag{15}$$

This inverse Bellman operator is obtained by solving the forward Bellman operator for $r(s, a)$ under the assumption that the value of absorbing states is zero. However, as pointed out by Kostrikov et al. (2019), such an assumption may introduce termination or survival bias; the value of absorbing states also needs to be learned. Our perspective provides a clear understanding of the effect of the inverse Bellman operator in Equation 15: The objective in Equation 10 will regress the $Q$-function of transitions into absorbing states towards $r_{\max}$ or $r_{\min}$, respectively. However, based on Equation 9, the implicit *reward* of absorbing states should be regressed toward $r_{\max}$ or $r_{\min}$.

Instead, we derive our inverse operator from the standard Bellman operator while exploiting that the value of the absorbing state $s_A$ is independent of the policy $\pi$

$$(\mathcal{T}_{\text{lsiq}}^{\pi}Q)(s,a) = Q(s,a) - \gamma \mathbb{E}_{s' \sim P(.|s,a)}\big((1-\nu)V^{\pi}(s') + \nu V(s_A)\big). \quad (16)$$

We further exploit that the value of the absorbing state can be computed in closed form as $V(s_A) = \frac{r_A}{1-\gamma}$, where $r_A$ equals $r_{\max}$ on expert states and $r_{\min}$ on policy states. Please note that the corresponding forward Bellman operator converges to the same $Q$-function, despite using the analytic value of absorbing states instead of bootstrapping, as we show in Appendix A.5. When applying our inverse operator in Equation 16 to Equation 9, we correctly regress the $Q$-function of transitions into absorbing states towards their discounted return. We show the resulting full objective in Appendix A.4.

We show the effect of our modified operator on the toy task depicted in Figure 1 (top), where the black point mass is spawned in either of the four dark blue squares and has to reach the green area in the middle. Once the agent enters the red area, the episode terminates. The expert always takes the shortest path to the green area, never visiting the red area. The operator proposed by IQ-Learn does not sufficiently penalize the agent for reaching absorbing states, preventing the IQ-Learn agent from reaching the goal consistently, as can be seen from the orange graph in Figure 1 (bottom). In contrast, when using our operator $\mathcal{T}_{\text{lsiq}}$, the agent solves the task successfully.

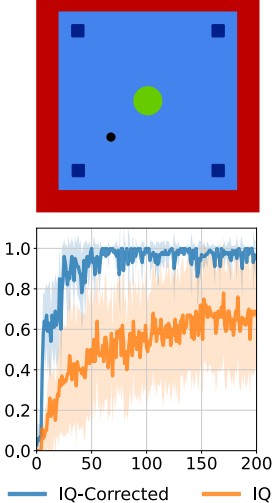

Figure 1: Point mass toy task (top) with success rate plot (bottom). Here, we compare the standard IQ-Learn operator to the modified operator.

### 3.5 An Alternative Formulation for the Expert Residual Minimization

The first term in Equation 9 defines the squared Bellman error minimization problem on the distribution $d_{\pi_E}$

$$\alpha\, \delta^2(d_{\pi_E}, r_{\max}) = \alpha \mathbb{E}_{d_{\pi_E}}\left[(r_Q(s,a) - r_{\max})^2\right]. \quad (17)$$

Due to bootstrapping, this minimization can become challenging, even for a fixed expert policy, as it does not fix the scale of the $Q$-function unless the trajectory reaches an absorbing state. This problem arises particularly on expert data for cyclic tasks, where we generate trajectories up to a fixed horizon. The lack of a fixed scale increases the variance of the algorithm, affecting the performance negatively.

Therefore, we propose a modified objective, analyzing Equation 17. The minimum of this term is achieved when $r_Q(s,a) = r_{\max}$ for all reachable $(s,a)$ under $d_{\pi_E}$. Thus, the objective of this term is to push the reward, on expert trajectories, towards $r_{\max}$. If we consider this minimum, each transition in the expert's trajectory has the following $Q$-value:

$$Q^{\pi_E}(s,a) = \sum_{t=0}^{\infty} \gamma^t r_{\max} = \frac{r_{\max}}{1-\gamma} = Q_{\max}, \quad \text{with } s,a \sim d_{\pi_E}(s,a). \quad (18)$$

As the objective of our maximization on expert distribution is equivalent to pushing the value of the expert's states and actions towards $Q_{\max}$, we propose to replace the bootstrapping target with the fixed target $Q_{\max}$ resulting in the following new objective:

$$\mathcal{L}_{\text{lsiq}}(Q) = \alpha \mathbb{E}_{d_{\pi_E}}\left[(Q(s,a) - Q_{\max})^2\right] + (1-\alpha)\mathbb{E}_{d_\pi}\left[\big(Q(s,a) - (r_{\min} + \gamma \mathbb{E}_{s' \sim P(.|s,a)}[V^*(s')])\big)^2\right]. \quad (19)$$

Note that we skip the terminal state treatment for clarity. The full objective is shown in Appendix A.4. Also, we omit the entropy term as we incorporate the latter now in $\mathcal{H}^{\pi}(s,a)$. This new objective incorporates a bias toward expert data. Therefore, it is not strictly equivalent to the original problem formulation. However, it updates the $Q$-function toward the same ideal target, while providing a simpler and more stable optimization landscape. Empirically, we experienced that this modification, while only justified intuitively, has a very positive impact on the algorithm's performance.

### 3.6 LEARNING FROM OBSERVATIONS

In many real-world tasks, we do not have access to expert actions, but only to observations of expert's behavior (Torabi et al., 2019b). In this scenario, AIL methods, such as GAIfO (Torabi et al., 2019a), can be easily adapted by learning a discriminator only depending on the current and the next state. Unfortunately, it is not straightforward to apply the same method to implicit rewards algorithms that learn a $Q$-function. The IQ-Learn method (Garg et al., 2021) relies on a simplification of the original objective to perform updates not using expert actions but rather actions sampled from the policy on expert states. However, this reformulation is not able to achieve good performance on standard benchmarks as shown in our experimental results.

A common practice used in the literature is to train an IDM. This approach has been previously used in behavioral cloning (Torabi et al., 2018; Nair et al., 2017) and for reinforcement learning from demonstrations (Guo et al., 2019; Pavse et al., 2020; Radosavovic et al., 2021). Following the same idea, we generate an observation-only version of our method by training an IDM online on policy data and using it for the prediction of unobserved actions of the expert. We modify the objective in Equation 19 to

$$\mathcal{L}_{\text{lsiq-o}}(Q) = \alpha \mathbb{E}_{d_{\pi_E}} \left[ \left( Q(s, \Gamma_\omega(s, s')) - Q_{\max} \right)^2 \right] + \bar{\alpha} \mathbb{E}_{d_\pi} \left[ \left( Q(s, a) - (r_{\min} + \gamma \mathbb{E}_{s' \sim P(\cdot|s, a)}[V^*(s')]) \right)^2 \right], \quad (20)$$

with the dynamics model $\Gamma_\omega(s, s')$, its parameters $\omega$ and $\bar{\alpha} = (1 - \alpha)$. We omit the notation for absorbing states and refer to Appendix A.4 instead. Notice that the IDM is only used to evaluate the expert actions, and is trained by solving the following optimization problem

$$\min_\omega \mathcal{L}_\Gamma(\omega) = \min_\omega \mathbb{E}_{d_\pi, \mathcal{P}} \left[ \|\Gamma_\omega(s, s') - a\|_2^2 \right], \quad (21)$$

where the expectation is performed on the state distribution generated by the learner policy $\pi$. While the mismatch between the training distribution and the evaluation distribution could potentially cause problems, our empirical evaluation shows that on the benchmarks we achieve performance similar to the action-aware algorithm. We give more details on this approach in Appendix B.

### 3.7 PRACTICAL ALGORITHM

We now instantiate a practical version of our algorithm in this section. An overview of our method is shown in Algorithm 1. In practice, we use parametric functions to approximate $Q$, $\pi$, $\mathcal{G}$ and $\Gamma$, and optimize the latter using gradient ascent on surrogate objective functions that approximate the expectations under $d_\pi$ and $d_{\pi_E}$ using the datasets $\mathcal{D}_\pi$ and $\mathcal{D}_{\pi_E}$. Further, we use target networks, as already suggested by the Garg et al. (2021). However, while the objective in Equation 4 lacked intuition about the usage of target networks, the objective in Equation 11 is equivalent to a reinforcement learning objective, in which target networks are a well-known tool for stabilization. Further, we exploit our access to the hard $Q$-function as well as our fixed reward target setting to

---

**Algorithm 1** LS-IQ

**Initialize:** $Q_\theta$, $\pi_\phi$, $\mathcal{G}_\zeta$ and optionally $\Gamma_\omega$
  **for** step $t$ in $\{1,...,N\}$ **do**
    Sample mini-batches $\mathcal{D}_\pi$ and $\mathcal{D}_{\pi_E}$
    (opt.) Predict actions for $\mathcal{D}_{\pi_E}$ using $\Gamma_\omega$
    $\mathcal{D}_{\pi_E} \leftarrow \{\{s, \Gamma_\omega(s, s'), s'\} | \forall \{s, s'\} \in \mathcal{D}_{\pi_E}\}$
    Update the $Q$-function using Eq. 20
    $\theta_{t+1} \leftarrow \theta_t + \kappa_Q \nabla_\theta [\mathcal{J}(\theta, \mathcal{D}_\pi, \mathcal{D}_{\pi_E})]$
    (opt.) Update $\mathcal{G}$-function using Eq. 14
    $\zeta_{t+1} \leftarrow \zeta_t - \kappa_\mathcal{G} \nabla_\zeta [\delta_\mathcal{G}^2(\zeta, \mathcal{D}_\pi)]$
    Update Policy $\pi_\phi$ using the KL
    $\phi_{t+1} \leftarrow \phi_t - \kappa_\pi \nabla_\phi [D_{KL}(\pi\phi \| \pi_{\tilde{Q}})]$
    (opt.) Update $\Gamma_\omega$ using Eq. 21
    $\omega_{t+1} \leftarrow \omega_t - \kappa_\Gamma \nabla_\omega [\mathcal{L}_\Gamma(\omega, \mathcal{D}_\pi)]$
  **end for**

---

calculate the maximum and minimum Q-values possible, $Q_{\min} = \frac{r_{\min}}{1 - \gamma}$ and $Q_{\max} = \frac{r_{\max}}{1 - \gamma}$, and clip the output of target network to that range. Note that this also holds for the absorbing states. In doing so, we ensure that the target $Q$ always remains in the desired range, which was often not the case with IQ-Learn. Target clipping prevents the explosion of the $Q$-values that can occur due to the use of neural approximators. This technique allows the algorithm to recover from poor value function estimates and prevents the $Q$-function from leaving the set of admissible functions. Finally, we found that training the policy on a small fixed expert dataset anneals the entropy bonus of expert trajectories, even if the policy never visits these states and actions. To address this problem, we clip the entropy bonus on expert states to a running average of the maximum entropy on policy states.

In continuous action spaces, $Z_s$ is intractable, which is why we can not directly extract the optimal policy using Equation 2. As done in previous work (Haarnoja et al., 2018; Garg et al., 2021), we use a parametric policy $\pi_\phi$ to approximate $\pi_{\tilde{Q}}$ by minimizing the KL $D_{\text{KL}}(\pi_\phi \| \pi_{\tilde{Q}})$. In our implementation, we found it unnecessary to use a double-critic update. This choice reduces the

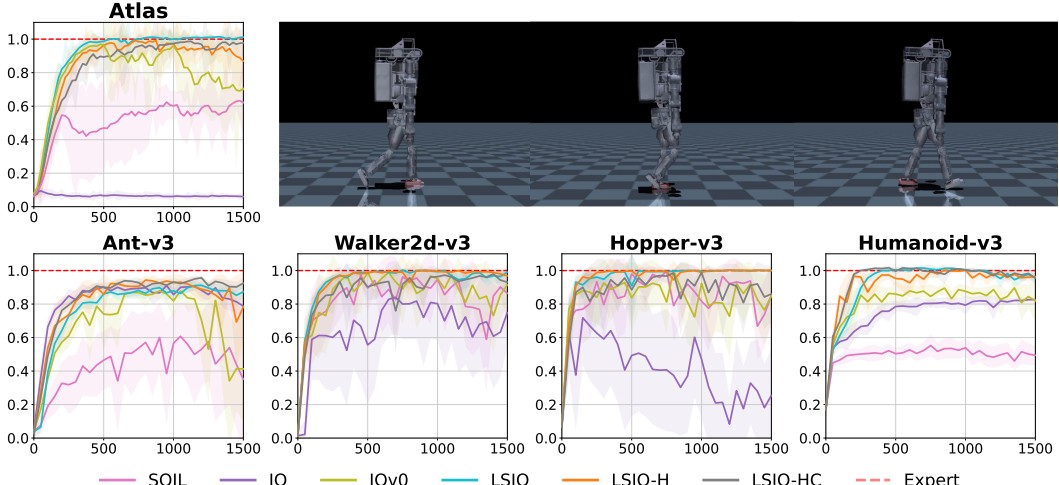

Figure 2: Comparison of different versions of LS-IQ. Abscissa shows the normalized discounted cumulative reward. Ordinate shows the number of training steps ($\times 10^3$). The first row shows the results and an exemplary trajectory – here the trained LS-IQ agent – on a locomotion task using an Atlas robot. The second row shows 4 MuJoCo Gym tasks, for which the expert's cumulative rewards are → Hopper:3299.81, Walker2d:5841.73, Ant:6399.04, Humanoid:6233.45

computational and memory requirements of the algorithm, making it comparable to SAC. Finally, we replace $V^*(s)$ with $V^\pi(s)$ on the policy expectation, as we do not have access to the latter in continuous action spaces.

## 4  EXPERIMENTS

We evaluate our method on six MuJoCo environments: Ant-v3, Walker2d-v3, Hopper-v3, HalfCheetah-v3, Humanoid-v3, and Atlas. The latter is a novel locomotion environment introduced by us and is further described in Appendix C.1. We select the following baselines: GAIL (Ho & Ermon, 2016), VAIL (Peng et al., 2019), IQ-Learn (Garg et al., 2021) and SQIL (Reddy et al., 2020). For a fair comparison, all methods are implemented in the same framework, Mushroom RL (D'Eramo et al., 2021). We verify that our implementations achieve comparable results to the original implementations by the authors. We use the hyperparameters proposed by the original authors for the respective environments and perform a grid search on novel environments. The original implementation of IQ-Learn evaluates two different algorithm variants depending on the given environment. We refer to these variants as IQv0—which uses telescoping (Garg et al., 2021) to evaluate the agent's expected return in Equation 4—, and IQ—which directly uses Equation 4— and evaluate both variants on all environments. For our method, we use the same hyperparameters as IQ-Learn, except for the regularizer coefficient $c$ and the entropy coefficient $\beta$, which we tune on each environment. We only consider equal mixing, i.e., $\alpha = 0.5$.

In our first experiment, we perform ablations on the different design choices of LSIQ. We evaluate the following variants: LSIQ-HC uses a (combined) entropy critic and regularization critic, LSIQ-H only uses the entropy critic, and LSIQ does not use any additional critic, similar to IQ-Learn. We use ten seeds and five expert trajectories for these experiments. For the Atlas environment, we use 100 trajectories. We also consider IQ, IQv0, and SQIL as baselines and show the learning curves for four environments in Figure 2. The learning curves on the HalfCheetah environment can be found in Appendix C.6. It is interesting to note that IQ-Learn without telescoping does not perform well on Atlas, Walker, and Hopper, where absorbing states are more likely compared to Ant and HalfCheetah, which almost always terminate after a fixed amount of steps. We hypothesize that the worse performance on Walker and Hopper is caused by reward bias, as absorbing states are not sufficiently penalized. IQv0 would suffer less from this problem as it treats all states visited by the agent as initial states, which results in stronger reward penalties for these states. We conduct further ablation studies showing the influence of the proposed techniques, including an ablation study on the effect of fixed targets, clipping on the target $Q$-value, entropy clipping for the expert, as well as

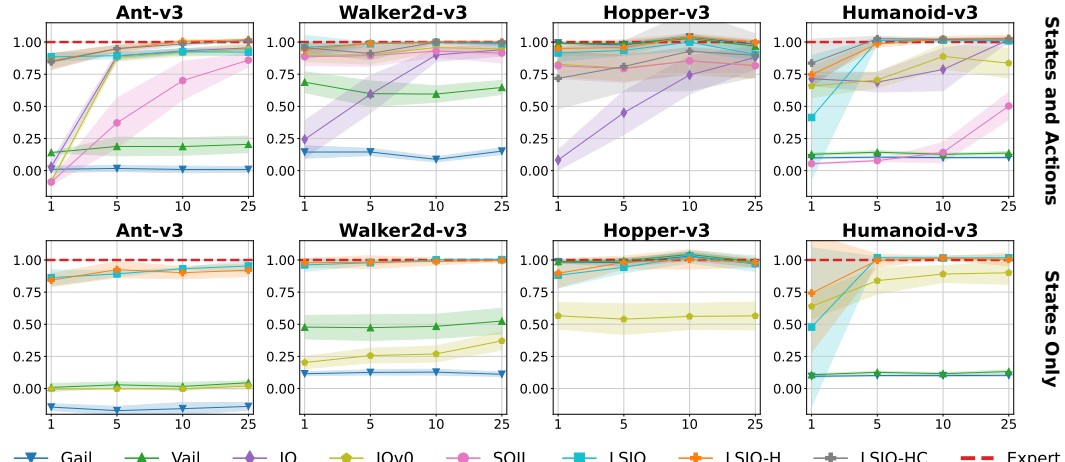

Figure 3: Ablation study on the effect of the number of expert trajectories on different Mujoco environments. Abscissa shows the normalized cumulative reward. Ordinate shows the number of expert trajectories. The first row shows the performance when considering states and action, while the second row considers the performance when using states only. Expert cumulative rewards identical to Figure 2.

the treatment of absorbing states in Appendix C. Our results show that the additional critics have little effect, while fixing the targets significantly increases the performance.

For our main experiments, we only evaluate LSIQ and LSIQ-H, which achieve the best performance in most environments. We compare our method to all baselines for four different numbers of expert demonstrations, 1, 5, 10, and 25, and always use five seeds. We perform each experiment with and without expert action. When actions are not available, we use a state transition discriminator (Torabi et al., 2019a) for GAIL and VAIL, and IDMs for LSIQ (c.f., Section 3.6). In contrast, IQ-Learn uses actions predicted on expert states by the current policy when no expert actions are available. In the learning-from-observation setting, we do not evaluate SQIL, and we omit the plots for IQ, which does not converge in any environment and focus only on IQv0. Figure 3 shows the final expected return over different numbers of demonstrations for four of the environments. All learning curves, including the HalfCheetah environment, can be found in Appendix C.6 for state-action setting and in Appendix C.5 for the learning-from-observation setting. Our experiments show that LSIQ achieves on-par or better performance compared to all baselines. In particular, in the learning-from-observation setting, LSIQ performs very well by achieving a similar return compared to the setting where states and actions are observed.

## 5 CONCLUSION

Inspired by the practical implementation of IQ-Learn, we derive a distribution matching algorithm using an implicit reward function and a squared $L_2$ penalty on the mixture distribution of the expert and the policy. We show that this regularizer minimizes a bounded $\chi^2$-divergence to the mixture distribution and results in modified updates for the $Q$-function and policy. Our analysis reveals an interesting connection to SQIL—which is not derived from an adversarial distribution matching objective—and shows that IQ-Learn suffers from reward bias. We build on our insights to propose a novel method, LS-IQ, which uses a modified inverse Bellman operator to address reward bias, target clipping, fixed reward targets for policy samples, and fixed $Q$-function targets for expert samples. We also show that the policy optimization of IQ-Learn is not consistent with regularization on the mixture distribution and show how this can be addressed by learning an additional regularization critic. In our experiments, LS-IQ outperforms strong baseline methods, particularly when learning from observations, where we train an IDM for predicting expert actions. In future work, we will quantify the bias introduced by the fixed $Q$-function target and investigate why this heuristic is fundamental for stabilizing learning. We will also analyze the error propagation in the $Q$-function target and derive theoretical guarantees on the $Q$-function approximation error.

ACKNOWLEDGMENTS

Calculations for this research were conducted on the Lichtenberg high-performance computer of the TU Darmstadt. This work was supported by the German Science Foundation (DFG) under grant number SE1042/41-1. Research presented in this paper has been partially supported by the German Federal Ministry of Education and Research (BMBF) within the subproject "Modeling and exploration of the operational area, design of the AI assistance as well as legal aspects of the use of technology" of the collaborative KIARA project (grant no. 13N16274).

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

## A  PROOFS AND DERIVATIONS

In this section, we present proofs of the propositions in the main paper. Furthermore, we provide two additional propositions on the optimal reward function and the bounds for the $\chi^2$-divergence when considering the mixture distribution.

### A.1  FROM OCCUPANCY MEASURES TO DISTRIBUTIONS

Based on Proposition A.1, the solution $\arg\max_{Q \in \tilde{\Omega}} \mathcal{J}_\rho(\tilde{Q}, \pi_{\tilde{Q}})$ under occupancy measures equals the solution $\arg\max_{\tilde{Q} \in \tilde{\Omega}} \mathcal{J}(\tilde{Q}, \pi_{\tilde{Q}})$ under state-action distributions. This result allows us to use the following distribution matching problem from now on:

$$\max_{\tilde{Q} \in \tilde{\Omega}} \mathcal{J}(\tilde{Q}, \pi_{\tilde{Q}}) = \max_{\tilde{Q} \in \tilde{\Omega}} \mathbb{E}_{d_{\pi_E}}\left[r_{\tilde{Q}}(s,a)\right] - \mathbb{E}_{d_{\pi_{\tilde{Q}}}}\left[r_{\tilde{Q}}(s,a)\right] - \psi(r_{\tilde{Q}}) - \beta H(\pi_{\tilde{Q}}), \tag{22}$$

where we introduce the implicit reward $r_{\tilde{Q}}(s,a) = \tilde{Q}(s,a) - \gamma \mathbb{E}_{s' \sim P(.|s,a)}[\tilde{V}^\pi(s')]$ for comprehension, $\beta$ is a constant controlling the entropy regularization, $d_{\pi_E}$ is the state-action distribution of the expert, and $d_\pi$ is the state-action distribution under the policy.

**Proposition A.1** *Let $\max_{r \in \mathcal{R}} \min_{\pi \in \Pi} L_\rho(\pi, r)$ be the dual problem of a regularized occupancy matching optimization problem and $L(\pi, r)$ be the Lagrangian of the regularized distribution matching problem. Then it holds that $L_\rho(\pi, r) \propto L(\pi, r)$ and $\mathcal{J}_\rho(\tilde{Q}, \pi_{\tilde{Q}}) \propto \mathcal{J}(\tilde{Q}, \pi_{\tilde{Q}})$.*

*Proof of Proposition A.1.*

Starting from the definition of the occupancy measure of an arbitrary policy $\pi$, we compute the normalizing constant as an integral:

$$\int_{\mathcal{S} \times \mathcal{A}} \rho_\pi(s,a) ds da = \int_{\mathcal{S} \times \mathcal{A}} \lim_{T \to \infty} \sum_{t=0}^{T-1} \gamma^t \mu_t^\pi(s) \pi(a|s) \, ds da$$

$$= \lim_{T \to \infty} \sum_{t=0}^{T-1} \gamma^t \int_{\mathcal{S} \times \mathcal{A}} \mu_t^\pi(s) \pi(a|s) \, ds da$$

$$= \lim_{T \to \infty} \sum_{t=0}^{T-1} \gamma^t \cdot 1$$

$$= \frac{1}{1-\gamma} \tag{23}$$

Now we compute the (discounted) state-action distribution as:

$$d_\pi(s,a) = \frac{\rho_\pi(s,a)}{\int_{\mathcal{S} \times \mathcal{A}} \rho(s,a) ds da} = \frac{\rho_\pi(s,a)}{\frac{1}{1-\gamma}} = (1-\gamma)\rho_\pi(s,a) \tag{24}$$

Thus, we have:

$$\rho_\pi(s,a) = \frac{1}{1-\gamma} d_\pi(s,a) \tag{25}$$

Using equation 25 in the definition of the objective we obtain:

$$\mathcal{J}(\pi) = \frac{1}{1-\gamma} \int_{\mathcal{S} \times \mathcal{A}} d_\pi(s,a) r(s,a) ds da = \frac{1}{1-\gamma} \mathbb{E}_{d_\pi}\left[r(s,a)\right] \tag{26}$$

A derivation similar to equation 26 can be done for the entropy and the regularizer using equation 25. Substituting the derived formulas into equation 1 and collecting the constant $\frac{1}{1-\gamma}$ proves the proposition. ∎

### A.2  THE BOUNDS OF THE $\chi^2$ DIVERGENCE ON MIXTURE DISTRIBUTIONS

**Proposition A.2** *Given the Pearson $\chi^2$-divergence between the distribution $d_{\pi_E}$ and the mixture distribution $0.5 \, d_{\pi_E} + 0.5 \, d_\pi$ in its variational form shown in Equation 6, the optimal reward is given by*

$$r^*(s,a) = \frac{1}{c} \frac{d_{\pi_E}(s,a) - d_\pi(s,a)}{d_{\pi_E}(s,a) + d_\pi(s,a)} \tag{27}$$

*and is bounded such that $r^*(s,a) \in [-1/c, 1/c]$ for all $s \in \mathcal{S}, a \in \mathcal{A}$.*

*Proof of Proposition A.2.* This proof follows the proof of Proposition 1 in Goodfellow et al. (2014). We recall that the $\chi^2$-divergence on a mixture in Equation 6 is

$$2\chi^2\Big(d_{\pi_E}\,\Big\|\,\underbrace{\tfrac{d_{\pi_E}+d_\pi}{2}}_{d_{\mathrm{mix}}}\Big)=\max_{r\in\mathcal{R}}2\Big(\mathbb{E}_{d_{\pi_E}}\big[r(s,a)\big]-\mathbb{E}_{d_{\mathrm{mix}}}\big[r(s,a)+kr(s,a)^2\big]\Big)$$

$$=\max_{r\in\mathcal{R}}\mathbb{E}_{d_{\pi_E}}\big[r(s,a)\big]-\mathbb{E}_{d_\pi}\big[r(s,a)\big]-k\mathbb{E}_{d_{\pi_E}}\big[r(s,a)^2\big]-k\mathbb{E}_{d_\pi}\big[r(s,a)^2\big]$$

$$=\max_{r\in\mathcal{R}}\int_{\mathcal{S}}\int_{\mathcal{A}}d_{\pi_E}(s,a)\big(r(s,a)-kr(s,a)^2\big)-d_\pi(s,a)\big(r(s,a)+kr(s,a)^2\big)\,da\,ds, \quad (28)$$

where $k=\,^1\!/_4$ for the conventional $\chi^2$-divergence. We generalize the $\chi^2$-divergence by setting $k=\,^c\!/_2$. For any $a,b\in\mathbb{R}^+\setminus\{0\}$, the function $y\to a(y-\tfrac{c}{2}\,y^2)-b(y+\tfrac{c}{2}\,y^2)$ achieves its maximum at $\tfrac{1}{c}\tfrac{a-b}{a+b}$, which belongs to the interval $[-\,^1\!/_c,\,^1\!/_c]$.

To conclude the proof, we notice that the reward function can be arbitrarily defined outside of $Supp(d_\pi)\cup Supp(d_{\pi_E})$, as it has no effect on the divergence. ∎

**Proposition A.3** *The Pearson $\chi^2$-divergence between the distribution $d_{\pi_E}$ and the mixture distribution $0.5\,d_{\pi_E}+0.5\,d_\pi$ is bounded as follows*

$$0\le 2\chi^2\big(d_{\pi_E}\,\big\|\,\underbrace{\tfrac{d_{\pi_E}+d_\pi}{2}}_{d_{mix}}\big)\le\tfrac{1}{c}\,. \quad (29)$$

*Proof of Proposition A.3.* To increase the readability, we drop the explicit dependencies on state and action in the notation, and we write $d_{\pi_E}$ and $d_\pi$ for $d_{\pi_E}(s,a)$ and $d_\pi(s,a)$, respectively. The lower bound is trivially true for any divergence and is reached when $d_\pi=d_{\mathrm{mix}}=d_{\pi_E}$. To prove the upper bound, we use the optimal reward function from Equation 27 and plug it into Equation 28 with $k=\,^c\!/_2$

$$2\chi^2\big(d_{\pi_E}\big\|\underbrace{\tfrac{d_{\pi_E}+d_\pi}{2}}_{d_{\mathrm{mix}}}\big)=\int_{\mathcal{S}}\int_{\mathcal{A}}d_{\pi_E}\big(r^*(s,a)-\tfrac{c}{2}r^*(s,a)^2\big)-d_\pi\big(r^*(s,a)+\tfrac{c}{2}r^*(s,a)^2\big)\,da\,ds$$

$$=\int_{\mathcal{S}}\int_{\mathcal{A}}d_{\pi_E}\left(\frac{1}{c}\left(\frac{d_{\pi_E}-d_\pi}{d_{\pi_E}+d_\pi}\right)-\frac{c}{2}\frac{1}{c^2}\left(\frac{d_{\pi_E}-d_\pi}{d_{\pi_E}+d_\pi}\right)^2\right)$$

$$-d_\pi\left(\frac{1}{c}\left(\frac{d_{\pi_E}-d_\pi}{d_{\pi_E}+d_\pi}\right)+\frac{c}{2}\frac{1}{c^2}\left(\frac{d_{\pi_E}-d_\pi}{d_{\pi_E}+d_\pi}\right)^2\right)da\,ds$$

$$=\int_{\mathcal{S}}\int_{\mathcal{A}}d_{\pi_E}\left(\frac{2d_{\pi_E}^2-2d_\pi^2-d_{\pi_E}^2+2d_{\pi_E}d_\pi-d_\pi^2}{2c(d_{\pi_E}+d_\pi)^2}\right)$$

$$-d_\pi\left(\frac{2d_{\pi_E}^2-2d_\pi^2+d_{\pi_E}^2-2d_{\pi_E}d_\pi+d_\pi^2}{2c(d_{\pi_E}+d_\pi)^2}\right)da\,ds$$

$$=\int_{\mathcal{S}}\int_{\mathcal{A}}\frac{d_{\pi_E}^3+d_\pi^3-d_{\pi_E}d_\pi^2-d_{\pi_E}^2d_\pi}{2c(d_{\pi_E}+d_\pi)^2}\,da\,ds=\frac{1}{2c}\int_{\mathcal{S}}\int_{\mathcal{A}}\frac{(d_{\pi_E}-d_\pi)^2}{d_{\pi_E}+d_\pi}\,da\,ds$$

$$=\frac{1}{2c}\int_{\mathcal{S}}\int_{\mathcal{A}}\frac{d_{\pi_E}^2}{d_{\pi_E}+d_\pi}+\frac{d_\pi^2}{d_{\pi_E}+d_\pi}-2\frac{d_{\pi_E}d_\pi}{d_{\pi_E}+d_\pi}\,da\,ds$$

$$=\frac{1}{2c}\left(\underbrace{\mathbb{E}_{d_{\pi_E}}\left[\frac{d_{\pi_E}}{d_{\pi_E}+d_\pi}\right]}_{\le 1}+\underbrace{\mathbb{E}_{d_\pi}\left[\frac{d_\pi}{d_{\pi_E}+d_\pi}\right]}_{\le 1}+\underbrace{\mathbb{E}_{d_{\pi_E}}\left[\frac{-2d_\pi}{d_{\pi_E}+d_\pi}\right]}_{\le 0}\right)\le\frac{1}{c}. \quad (30)$$

Note that the bound is tight, as the individual bounds of each expectation are only achieved in conjunction. ∎

**Proposition A.4** *Let $\chi^2\big(d_{\pi_E}\,\big\|\,\alpha d_{\pi_E}+(1-\alpha)d_\pi\big)$ be the Pearson $\chi^2$-divergence between the distribution $d_{\pi_E}$ and the mixture distribution $\alpha d_{\pi_E}+(1-\alpha)d_\pi$. Then it holds that:*

$$\chi^2\big(d_{\pi_E}\,\big\|\,\alpha d_{\pi_E}+(1-\alpha)d_\pi\big)\le(1-\alpha)\chi^2\big(d_{\pi_E}\,\big\|\,d_\pi\big)\,.$$

*Proof of Proposition A.4.* The proof follows straightforwardly from the joint convexity of $f$-divergences:

$$D_f\big(\kappa P_1+(1-\kappa)P_2\,\big\|\,\kappa Q_1+(1-\kappa)Q_2\big)\le\kappa D_f(P_1\,\|\,Q_1)+(1-\kappa)D_f(P_2\,\|\,Q_2)\,,$$

where $\kappa \in [0,1]$ is the a mixing coefficient. The $\chi^2$-divergence is an $f$-divergence with $f(t) = (t-1)^2$. Now using the $\chi^2$-divergence, and let $P = P_1 = P_2 = Q_2$, $D = Q_1$, and $\alpha = (1-\kappa)$:

$$\chi^2\left(P \parallel \alpha P + (1-\alpha)D\right) \leq (1-\alpha)\,\chi^2\left(P \parallel D\right) + \underbrace{\alpha\chi^2\left(P \parallel P\right)}_{=0}$$

$$\chi^2\left(P \parallel \alpha P + (1-\alpha)D\right) \leq (1-\alpha)\,\chi^2\left(P \parallel D\right).$$

Setting $P = d_{\pi_E}$ and $D = d_\pi$ concludes the proof. ∎

## A.3 From $\chi^2$-regularized MaxEnt-IRL to Least-Squares Reward Regression

We recall that the entropy-regularized IRL under occupancy measures is given by

$$L_\rho(r,\pi) = (-\beta H_\rho(\pi) - \mathbb{E}_{\rho_\pi}[r(s,a)]) + \mathbb{E}_{\rho_{\pi_E}}[r(s,a)] - \psi_\rho(r). \tag{31}$$

We also recall that using soft inverse Bellman operator to do the change of variable yields:

$$\mathcal{J}_\rho(\tilde{Q}, \pi_{\tilde{Q}}) = \mathbb{E}_{\rho_{\pi_E}}\left[\tilde{Q}(s,a) - \gamma \mathbb{E}_{s' \sim P(.|s,a)}[\tilde{V}^\pi(s')]\right] - \beta H_\rho(\pi_{\tilde{Q}}) \tag{32}$$
$$- \mathbb{E}_{\rho_\pi}\left[\tilde{Q}(s,a) - \gamma \mathbb{E}_{s' \sim P(.|s,a)}[\tilde{V}^\pi(s')]\right] - \psi_\rho(r).$$

We can now use Proposition A.1 to switch the objective from occupancy measures to distributions.

*Proof of Proposition 3.1.* Staring from the objective function in Equation 22, by expanding the expectation and rearranging the terms, we obtain:

$$\mathcal{J}(\tilde{Q}, \pi_{\tilde{Q}}) = \mathbb{E}_{d_{\pi_E}}\left[r_{\tilde{Q}}(s,a)\right] - \mathbb{E}_{d_\pi}\left[r_{\tilde{Q}}(s,a)\right] - c\,\mathbb{E}_{d_{\tilde{d}}}\left[r_{\tilde{Q}}(s,a)^2\right] - \beta H(\pi_{\tilde{Q}})$$

$$= \mathbb{E}_{d_{\pi_E}}\left[r_{\tilde{Q}}(s,a)\right] - \mathbb{E}_{d_\pi}\left[r_{\tilde{Q}}(s,a)\right]$$
$$- c\alpha\,\mathbb{E}_{d_{\pi_E}}\left[r_{\tilde{Q}}(s,a)^2\right] - c(1-\alpha)\,\mathbb{E}_{d_\pi}\left[r_{\tilde{Q}}(s,a)^2\right] - \beta H(\pi_{\tilde{Q}})$$

$$= -\mathbb{E}_{d_{\pi_E}}\left[c\alpha\,r_{\tilde{Q}}(s,a)^2 - r_{\tilde{Q}}(s,a)\right] - \mathbb{E}_{d_\pi}\left[c(1-\alpha)\,r_{\tilde{Q}}(s,a)^2 + r_{\tilde{Q}}(s,a)\right] - \beta H(\pi_{\tilde{Q}})$$

$$= -c\left(\alpha\,\mathbb{E}_{d_{\pi_E}}\left[r_{\tilde{Q}}(s,a)^2 - \tfrac{1}{\alpha c}r_{\tilde{Q}}(s,a)\right]\right.$$
$$\left. + (1-\alpha)\,\mathbb{E}_{d_\pi}\left[r_{\tilde{Q}}(s,a)^2 + \tfrac{1}{(1-\alpha)c}r_{\tilde{Q}}(s,a)\right] + \tfrac{\beta}{c}H(\pi_{\tilde{Q}})\right)$$

Defining $r_{\max} = \frac{1}{2\alpha c}$ and $r_{\min} = -\frac{1}{2(1-\alpha)c}$ and completing the squares we obtain:

$$\mathcal{J}(\tilde{Q}, \pi_{\tilde{Q}}) = -c\left(\alpha\,\mathbb{E}_{d_{\pi_E}}\left[r_{\tilde{Q}}(s,a)^2 - \tfrac{1}{\alpha c}r_{\tilde{Q}}(s,a)\right] + (1-\alpha)\,\mathbb{E}_{d_\pi}\left[r_{\tilde{Q}}(s,a)^2 + \tfrac{1}{(1-\alpha)c}r_{\tilde{Q}}(s,a)\right]\right.$$
$$\left. + \tfrac{\beta}{c}H(\pi_{\tilde{Q}})\right) + \alpha c\,(r_{\max}^2 - r_{\max}^2) + (1-\alpha)c\,(r_{\min}^2 - r_{\min}^2)$$

$$= -c\left(\alpha\,\mathbb{E}_{d_{\pi_E}}\left[r_{\tilde{Q}}(s,a)^2 - \tfrac{1}{\alpha c}r_{\tilde{Q}}(s,a) + r_{\max}^2\right] + (1-\alpha)\,\mathbb{E}_{d_\pi}\left[r_{\tilde{Q}}(s,a)^2\right.\right.$$
$$\left.\left. + \tfrac{1}{(1-\alpha)c}r_{\tilde{Q}}(s,a) + r_{\min}^2\right] + \tfrac{\beta}{c}H(\pi_{\tilde{Q}})\right) + \alpha c\,r_{\max}^2 + (1-\alpha)c\,r_{\min}^2$$

$$= -c\left(\alpha\,\mathbb{E}_{d_{\pi_E}}\left[\left(r_{\tilde{Q}}(s,a) - \tfrac{1}{2\alpha c}\right)^2\right] + (1-\alpha)\,\mathbb{E}_{d_\pi}\left[\left(r_{\tilde{Q}}(s,a) + \tfrac{1}{2(1-\alpha)c}\right)^2\right]\right.$$
$$\left. + \tfrac{\beta}{c}H(\pi_{\tilde{Q}})\right) + \alpha c\,r_{\max}^2 + (1-\alpha)c\,r_{\min}^2.$$

Finally, we obtain the following result:

$$\mathcal{J}(\tilde{Q}, \pi_{\tilde{Q}}) = -c\left(\alpha\,\mathbb{E}_{d_{\pi_E}}\left[\left(r_{\tilde{Q}}(s,a) - r_{\max}\right)^2\right] + (1-\alpha)\,\mathbb{E}_{d_\pi}\left[\left(r_{\tilde{Q}}(s,a) - r_{\min}\right)^2\right] + \tfrac{\beta}{c}H(\pi_{\tilde{Q}})\right) + K, \tag{33}$$

where $K = \alpha c\,r_{\max}^2 + (1-\alpha)c\,r_{\min}^2 = \frac{1}{4\alpha c} + \frac{1}{4(1-\alpha)c}$ is a fixed constant.

Comparing Equation 9 with Equation 33 results in

$$\mathcal{J}(\tilde{Q}, \pi_{\tilde{Q}}) + K \propto \mathcal{L}(\tilde{Q}, \pi_{\tilde{Q}}). \tag{34}$$

Given that an affine transformation (with positive multiplicative constants) preserves the optimum, $\arg\max_{\tilde{Q} \in \tilde{\Omega}} \bar{\mathcal{J}}(\tilde{Q}, \pi_{\tilde{Q}})$ is the solution of the entropy-regularized least squares objective $\mathcal{L}(\tilde{Q}, \pi_{\tilde{Q}})$.

∎

### A.4 FULL LS-IQ OBJECTIVE WITH TERMINAL STATES HANDLING

Inserting our inverse Bellman operator derived in Section 3.4 into the least-squares objective defined in Equation 9 and rearranging the terms yields the following objective for hard $Q$-functions:

$$
\begin{aligned}
\mathcal{L}(Q, \pi_Q) = {}& \alpha \mathbb{E}_{\substack{s,a \sim d_{\pi_E} \\ s' \sim P(.|s,a)}} \left[ (1 - \nu) \left( Q(s,a) - (r_{\max} + \gamma V^\pi(s')) \right)^2 \right] \\
& + \alpha \mathbb{E}_{\substack{s,a \sim d_{\pi_E} \\ s' \sim P(.|s,a)}} \left[ \nu \left( Q(s,a) - (r_{\max} + \gamma \tfrac{r_{\max}}{1-\gamma}) \right)^2 \right] \\
& + (1 - \alpha) \mathbb{E}_{\substack{s,a \sim d_{\pi_Q} \\ s' \sim P(.|s,a)}} \left[ (1 - \nu) \left( Q(s,a) - (r_{\min} + \gamma V^\pi(s')) \right)^2 \right] \\
& + (1 - \alpha) \mathbb{E}_{\substack{s,a \sim d_{\pi_Q} \\ s' \sim P(.|s,a)}} \left[ \nu \left( Q(s,a) - (r_{\min} + \gamma \tfrac{r_{\min}}{1-\gamma}) \right)^2 \right] + \frac{\beta}{c} H(\pi_Q)
\end{aligned}
\tag{35}
$$

$$
\begin{aligned}
\mathcal{L}(Q, \pi_Q) = {}& \alpha \mathbb{E}_{\substack{s,a \sim d_{\pi_E} \\ s' \sim P(.|s,a)}} \left[ (1 - \nu) \left( Q(s,a) - (r_{\max} + \gamma V^\pi(s')) \right)^2 \right] \\
& + \alpha \mathbb{E}_{\substack{s,a \sim d_{\pi_E} \\ s' \sim P(.|s,a)}} \left[ \nu \left( Q(s,a) - \tfrac{r_{\max}}{1-\gamma} \right)^2 \right] \\
& + (1 - \alpha) \mathbb{E}_{\substack{s,a \sim d_{\pi_Q} \\ s' \sim P(.|s,a)}} \left[ (1 - \nu) \left( Q(s,a) - (r_{\min} + \gamma V^\pi(s')) \right)^2 \right] \\
& + (1 - \alpha) \mathbb{E}_{\substack{s,a \sim d_{\pi_Q} \\ s' \sim P(.|s,a)}} \left[ \nu \left( Q(s,a) - \tfrac{r_{\min}}{1-\gamma} \right)^2 \right] + \frac{\beta}{c} H(\pi_Q) .
\end{aligned}
\tag{36}
$$

Now including the fixed target for the expert distribution introduced in Section 3.5 yields:

$$
\begin{aligned}
\mathcal{L}_{\text{lsiq}}(Q, \pi_Q) = {}& \alpha \mathbb{E}_{\substack{s,a \sim d_{\pi_E} \\ s' \sim P(.|s,a)}} \left[ (1 - \nu) \left( Q(s,a) - \tfrac{r_{\max}}{1-\gamma} \right)^2 \right] \\
& + \alpha \mathbb{E}_{\substack{s,a \sim d_{\pi_E} \\ s' \sim P(.|s,a)}} \left[ \nu \left( Q(s,a) - \tfrac{r_{\max}}{1-\gamma} \right)^2 \right] \\
& + (1 - \alpha) \mathbb{E}_{\substack{s,a \sim d_{\pi_Q} \\ s' \sim P(.|s,a)}} \left[ (1 - \nu) \left( Q(s,a) - (r_{\min} + \gamma V^\pi(s')) \right)^2 \right] \\
& + (1 - \alpha) \mathbb{E}_{\substack{s,a \sim d_{\pi_Q} \\ s' \sim P(.|s,a)}} \left[ \nu \left( Q(s,a) - \tfrac{r_{\min}}{1-\gamma} \right)^2 \right] + \frac{\beta}{c} H(\pi_Q)
\end{aligned}
\tag{37}
$$

$$
\begin{aligned}
\mathcal{L}_{\text{lsiq}}(Q, \pi_Q) = {}& \alpha \mathbb{E}_{\substack{s,a \sim d_{\pi_E} \\ s' \sim P(.|s,a)}} \left[ \left( Q(s,a) - \tfrac{r_{\max}}{1-\gamma} \right)^2 \right] \\
& + (1 - \alpha) \mathbb{E}_{\substack{s,a \sim d_{\pi_Q} \\ s' \sim P(.|s,a)}} \left[ (1 - \nu) \left( Q(s,a) - (r_{\min} + \gamma V^\pi(s')) \right)^2 \right] \\
& + (1 - \alpha) \mathbb{E}_{\substack{s,a \sim d_{\pi_Q} \\ s' \sim P(.|s,a)}} \left[ \nu \left( Q(s,a) - \tfrac{r_{\min}}{1-\gamma} \right)^2 \right] + \frac{\beta}{c} H(\pi_Q) ,
\end{aligned}
\tag{38}
$$

where Equation 38 is the full LS-IQ objective for our hard $Q$-function. For the observations-only setting we predict the expert's actions using the IDM.

### A.5 CONVERGENCE OF OUR FORWARD BACKUP OPERATOR

As described in Section 3.4, our inverse operator,

$$
(\mathcal{T}^\pi_{\text{lsiq}} Q)(s,a) = Q(s,a) - \gamma \mathbb{E}_{s' \sim P(.|s,a)} \left( (1 - \nu) V^\pi(s') + \nu V(s_A) \right),
\tag{39}
$$

is based on the standard Bellman backup operator, except that, instead of bootstrapping, we use the known values for transitions into absorbing state. We will now show that repeatedly applying the corresponding forward Operator

$$
(\mathcal{B}^\pi_{\text{lsiq}} Q)(s,a) = r(s,a) + \gamma \mathbb{E}_{s' \sim P(.|s,a)} \left( (1 - \nu) V^\pi(s') + \nu V(s_A) \right),
\tag{40}
$$

converges to the Q function. Our proof is based on the same technique that is commonly used to prove convergence of the standard Bellmen operator, namely by showing that the Q function

$Q^\pi(s, a) := r(s, a) + \gamma E_{s' \sim p(s'|s,a)} \mathbb{E}_{a' \sim \pi(a'|s')} Q^\pi(s', a')$ is a fixed point of our operator, that is, $(\mathcal{B}^\pi_{\text{lsiq}} Q^\pi)(s, a) = Q^\pi(s, a)$, and by further showing that our operator is a contraction,

$$||(\mathcal{B}^\pi_{\text{lsiq}} Q_A)(s, a) - (\mathcal{B}^\pi_{\text{lsiq}} Q_B)(s, a)||_\infty \leq \gamma ||Q_A(s, a) - Q_B(s, a)||_\infty, \tag{41}$$

were $||.||_\infty$ is the maximum norm; here, we assume a finite states and actions for the sake of simplicity.

**Proposition A.5** *The Q function of policy $\pi$ is a fixed point of $\mathcal{B}^\pi_{\text{lsiq}}$,*

$$(\mathcal{B}^\pi_{\text{lsiq}} Q^\pi(s, a)) = Q^\pi(s, a). \tag{42}$$

*Proof of Proposition A.5.* The proof follows straightforwardly from the fact that our forward operator performs the same update as the standard Bellman operator, if applied to the actual Q function of the policy, $Q^\pi(s, a)$, since then $V(s_A) := \frac{r(s_A)}{(1-\gamma)} = \mathbb{E}_{a' \sim \pi(.|_A)} Q^\pi(s', a'))$. Thus,

$$(\mathcal{B}^\pi_{\text{lsiq}} Q^\pi(s, a)) = r(s, a) + \gamma \mathbb{E}_{s' \sim P(.|s,a)} \left( (1 - \nu) V^\pi(s') + \nu V(s_A) \right) \tag{43}$$

$$= r(s, a) + \gamma \mathbb{E}_{s' \sim P(.|s,a)} \left( (1 - \nu) \mathbb{E}_{a' \sim \pi(a'|s')} Q^\pi(s', a') + \nu \mathbb{E}_{a' \sim \pi(.|s')} Q^\pi(s', a') \right) \tag{44}$$

$$= r(s, a) + \gamma \mathbb{E}_{s' \sim P(.|s,a)} \mathbb{E}_{a' \sim \pi(.|s')} Q^\pi(s', a') = Q^\pi(s, a). \tag{45}$$

∎

**Proposition A.6** *The forward operator $\mathcal{B}^\pi_{\text{lsiq}}$ is a contraction,*

$$||(\mathcal{B}^\pi_{\text{lsiq}} Q_A)(s, a) - (\mathcal{B}^\pi_{\text{lsiq}} Q_B)(s, a)||_\infty \leq \gamma ||Q_A(s, a) - Q_B(s, a)||_\infty. \tag{46}$$

*Proof of Proposition A.6.*

$$\left\| (\mathcal{B}^\pi_{\text{lsiq}} Q_A)(s, a) - (\mathcal{B}^\pi_{\text{lsiq}} Q_B)(s, a) \right\|_\infty \tag{47}$$

$$= \max_{s,a} \left| \gamma \mathbb{E}_{s' \sim P(.|s,a)} \mathbb{E}_{a' \sim \pi(a'|s')} \left[ (1 - \nu)(Q_A(s', a') - Q_B(s', a')) \right] \right| \tag{48}$$

$$\leq \gamma \max_{s,a} \left| \mathbb{E}_{s' \sim P(.|s,a)} \mathbb{E}_{a' \sim \pi(a'|s')} \left[ (Q_A(s', a') - Q_B(s', a')) \right] \right| \tag{49}$$

$$\leq \gamma \max_{s',a'} \left| Q_A(s', a') - Q_B(s', a') \right| \tag{50}$$

$$= \gamma \left\| Q_A(s, a) - Q_B(s, a) \right\|_\infty \tag{51}$$

∎

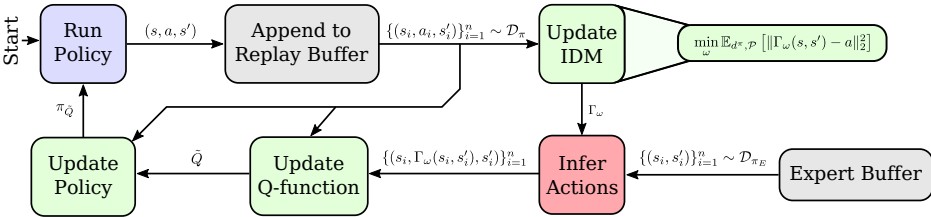

Figure 4: Training procedure of the IDM in LS-IQ.

## B  LEARNING FROM OBSERVATIONS

This section describes the IDM in greater detail. Figure 4 illustrates the training procedure of the IDM in LS-IQ. As can be seen, the IDM uses the replay buffer data generated by an agent to infer the actions from state transitions. Therefore, the simple regression loss from Equation 21 is used. At the beginning of training, the IDM learns on transitions generated by a (random) agent. Once the agent gets closer to the expert distribution, the IDM is trained on transitions closer to the expert. The intuition behind the usage of an IDM arises from the idea that, while two agents might produce different trajectories and, consequently, state-action distributions, the underlying dynamics are shared. This allows the IDM to infer more information about an action corresponding to a state transition than a random action predicted by the policy includes, as done by Garg et al. (2021). The experiments in Section 4 show that using an IDM yields superior or on-par performance w.r.t. the baselines in the state-only scenario.

While we only present a deterministic IDM, we also experimented with stochastic ones. For instance, we modeled the IDM as a Gaussian distribution and trained it using a maximum likelihood loss. We also tried a fully Bayesian approach to impose a prior, where we learned the parameters of a Normal-Inverse Gamma distribution and used a Student's t distribution for predicting actions of state transitions, as done by Amini et al. (2020). However, stochastic approaches did not show any notable benefit, therefore we stick to the simple deterministic approach.

## C  EXPERIMENTS

This section contains environment descriptions and additional results that have been omitted in the main paper due to space constraints.

### C.1  THE ATLAS LOCOMOTION ENVIRONMENT

The Atlas locomotion environment is a novel locomotion environment introduced by us. This environment aims to train agents on more realistic tasks, in contrast to the Mujoco Gym tasks, which generally have fine-tuned dynamics explicitly targeted towards reinforcement learning agents. The Atlas environment fixes the arms by default, resulting in 10 active joints. Each joint is torque-controlled by one motor. The state space includes all joint positions and velocities as well as 3D forces on the front and back foot, yielding a state-dimensionality of $D_s = 20 + 2 \cdot 2 \cdot 3 = 32$. The action space includes the desired torques for each joint motor, yielding an action dimensionality of $D_a = 10$. Optionally, the upper body with the arms can be activated, extending the number of joints and actuators to 27. The Atlas environment is implemented using Mushroom-RL's (D'Eramo et al., 2021) Mujoco interface.

For the sake of completeness, we added the cumulative reward plots – in contrast to the *discounted* cumulative reward plots as in Figure 2 – with an additional VAIL agent in Figure 5. The reward used as a metric for the performance is defined as $r = \exp(-(v_\pi - v_{\pi_E})^2)$, where $v_\pi$ is the agent's upper body velocity and $v_{\pi_E}$ is the expert's upper body velocity. The expert's walking velocity is $1.25\frac{m}{s}$. The code for the environment as well as the expert data is available at https://github.com/robfiras/ls-iq.

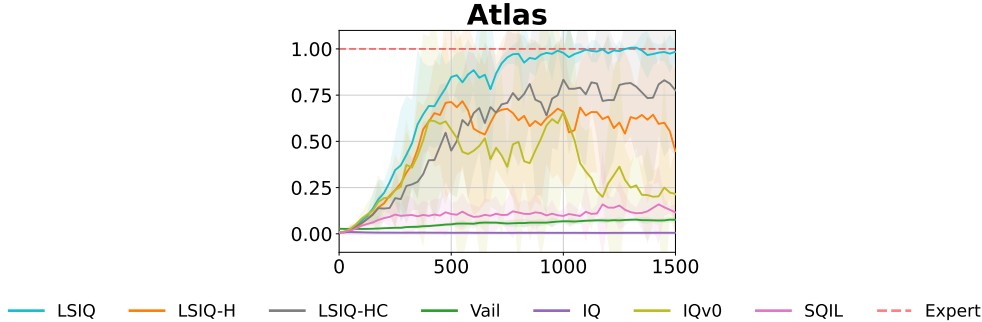

Figure 5: Training results and an exemplary trajectory – here the trained LSIQ agent – of a locomotion task using as simulated Atlas robot. Abscissa shows the normalized cumulative reward. Ordinate shows the number of training steps ($\times 10^3$).

## C.2 ABLATION STUDY: ABSORBING STATE TREATMENT AND TARGET $Q$ CLIPPING

Figure 6 presents ablations on the effects of the proposed treatment of absorbing states and the clipping of the $Q$-function target on an LSIQ agent with bootstrapping. To see the pure effect of the absorbing state treatment and the clipping, we did not include fixed targets. Note that the fixed target implicitly treats absorbing states of the expert, as it provides the same target for states and actions transitioning towards absorbing states. We have chosen a LSIQ agent without an entropy and regularization critic. The experiments are conducted on the Humanoid-v3 task, as the tasks HalfCheetah-v3, Ant-v3, either do not have or have very rare absorbing states, and Walker-v3 and Hopper-v3 are too easy to see the effects. We use a regularizer constant of $c = 0.5$ and a mixing parameter of $\alpha = 0.5$ yielding a maximum reward of $r_{\max} = \frac{1}{2(1-0.5)0.5} = 2$ and a minimum reward of $r_{\min} = -\frac{1}{2(1-0.5)0.5} = -2$. This yields a maximum $Q$-value of $Q_{\max} = \frac{r_{\max}}{1-\gamma} = 200$ and a minimum $Q$-value of $Q_{\min} = \frac{r_{\min}}{1-\gamma} = -200$. The rows show the different agent configurations: First, LSIQ with clipping and absorbing state treatment; second, LSIQ with clipping but no absorbing state treatment; and lastly, LSIQ without clipping and no treatment of absorbing states – which is equivalent to SQIL with symmetric reward targets. For a better understanding of the effect, the plots show the individual seeds of a configuration. As can be seen, the first LSIQ agent can successfully learn the task and regresses the $Q$-value of the transitions towards absorbing states to the minimum $Q$-value of -200. The second configuration does not treat absorbing states and is not able to learn the task with all seeds. As can be seen, the average $Q$-value of absorbing states is between -2 and -6. Taking a closer look at the first two plots in the second row, one can see that those seeds did not learn, whose average $Q$-value on non-absorbing transitions is close or even below the average $Q$-values of states and actions yielding to absorbing states. This strengthens the importance of our terminal state treatment, which pulls $Q$-values of states and action towards absorbing states to the lowest possible $Q$-value and, therefore, avoids a termination bias. Finally, one can see the problem of exploding $Q$-value in the last LSIQ configuration. This is evident by the scale of the abscissa, highlighted in the plots. Interestingly, while some seeds still perform reasonably well despite the enormously high $Q$-value, it clearly correlates to the high variance in the cumulative reward plot.

## C.3 ABLATION STUDY: INFLUENCE OF FIXED TARGETS AND ENTROPY CLIPPING

To show the effect of the fixed target (c.d., Section 3.5) and the entropy clipping (c.f., 3.7), we conducted a range of ablation studies for different versions of LSIQ on all Mujoco task. The results are shown in Figure 7 for the LSIQ version only with an entropy critic and in Figure 8 for the LSIQ version with an entropy and a regularization critic. As can be seen from the Figures, the version with the fixed target and the entropy clipping performs at best. It is especially noteworthy that the entropy clipping becomes of particular importance on tasks that require a high temperature parameter $\beta$, which is the case for the Humanoid-v3 environment.

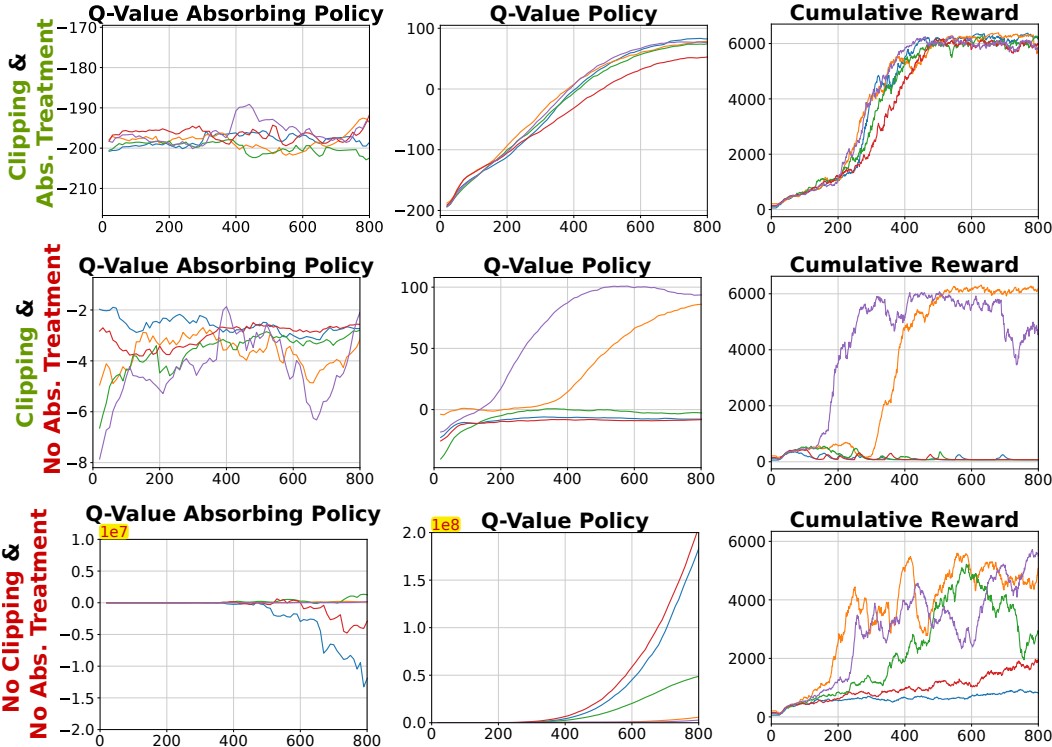

Figure 6: Ablation study on the effect of the proposed treatment of absorbing states and the clipping of the $Q$-value target on a LSIQ agent with bootstrapping (no fixed targets). The experiments are conducted on the Humanoid-v3 task, with an expert reaching a cumulative reward of 6233.45. Multiple lines in each plot show the **individual seeds**. The first column presents the average $Q$-value of states and actions **yielding to an absorbing state** visited by the policy. The second column presents the average $Q$-value of all states and actions that do not yield in absorbing states visited by the policy. The third column presents the cumulative reward. The rows present the ablations done to the LSIQ agent. Ordinate shows the number of training steps ($\times 10^3$).

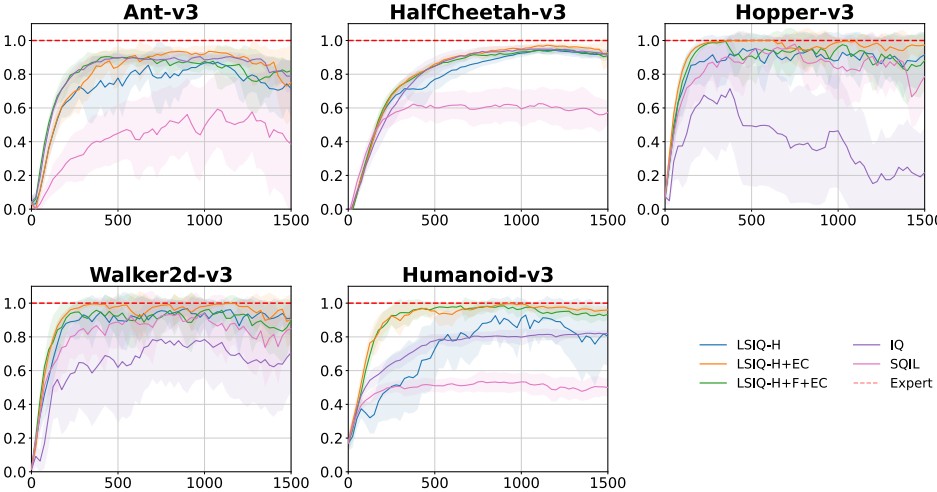

Figure 7: Comparison of different versions of **LSIQ-H** (**Regularization Critic**): First, the bootstrapping version → LSIQ-H; second, the bootstrapping version with entropy clipping → LSIQ-H+EC; thirdly, the fixed target version with entropy clipping → LSIQ-H+EC+FT. IQ and SQIL are added for reference. Abscissa shows the normalized discounted cumulative reward. Ordinate shows the number of training steps ($\times 10^3$). Experiments are conducted with **5** expert trajectories and five seeds. Expert cumulative rewards → Hopper:3299.81, Walker2d:5841.73, Ant:6399.04, HalfCheetah:12328.78, Humanoid:6233.45

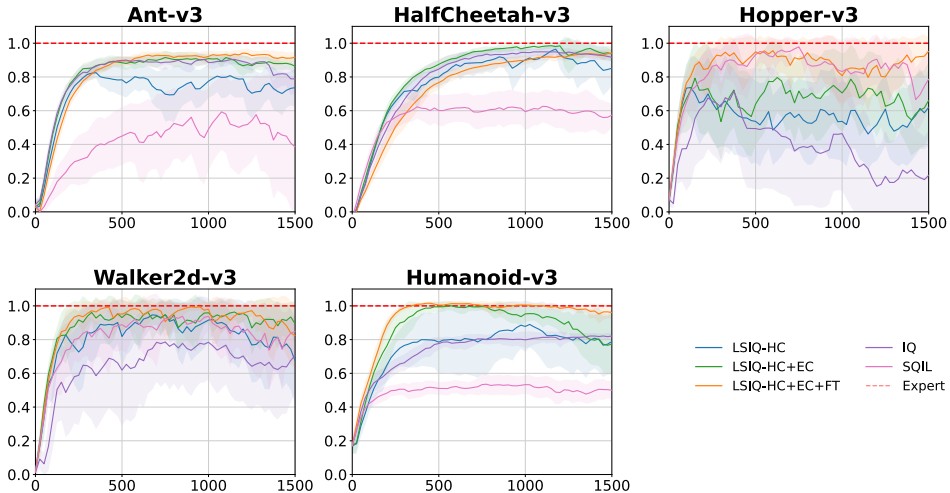

Figure 8: Comparison of different versions of **LSIQ-HC** (**Entropy+Regularization Critic**): First, the bootstrapping version → LSIQ-HC; second, the bootstrapping version with entropy clipping → LSIQ-HC+EC; thirdly, the fixed target version with entropy clipping → LSIQ-HC+EC+FT. IQ and SQIL are added for reference. Abscissa shows the normalized discounted cumulative reward. Ordinate shows the number of training steps ($\times 10^3$). Experiments are conducted with **5** expert trajectories and five seeds. Expert cumulative rewards → Hopper:3299.81, Walker2d:5841.73, Ant:6399.04, HalfCheetah:12328.78, Humanoid:6233.45

## C.4 ALL EXPERIMENT RESULTS OF THE DIFFERENT VERSION OF LSIQ

Figure 9 presents all the plots presented in Figure 2 with the additional Humanoid-v3 results. Figure 10 and Figure 11 correspond to Figure 3 but show all five environments. The corresponding learning curves are shown in Appendix C.6 and C.5.

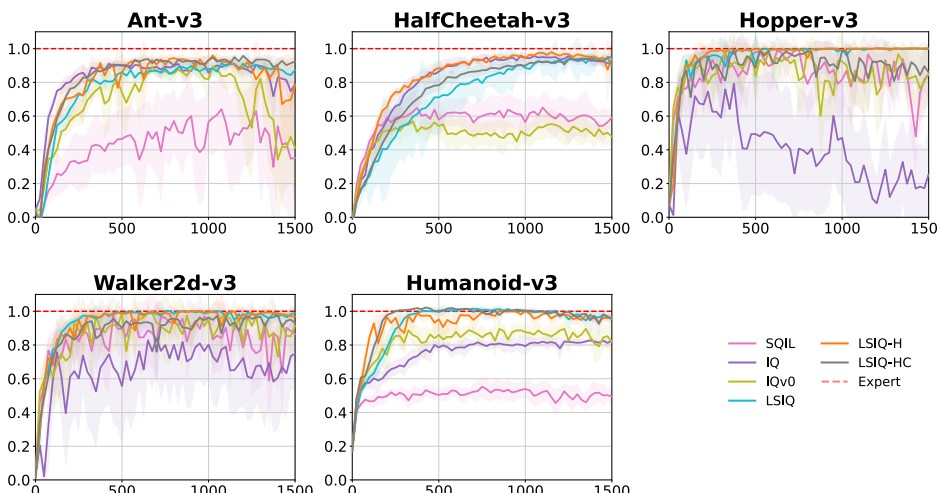

Figure 9: Comparison of different versions of LS-IQ. Now also with the Humanoid-v3 environment. Abscissa shows the normalized discounted cumulative reward. Ordinate shows the number of training steps ($\times 10^3$). Expert cumulative rewards → Hopper:3299.81, Walker2d:5841.73, Ant:6399.04, HalfCheetah:12328.78, Humanoid:6233.45

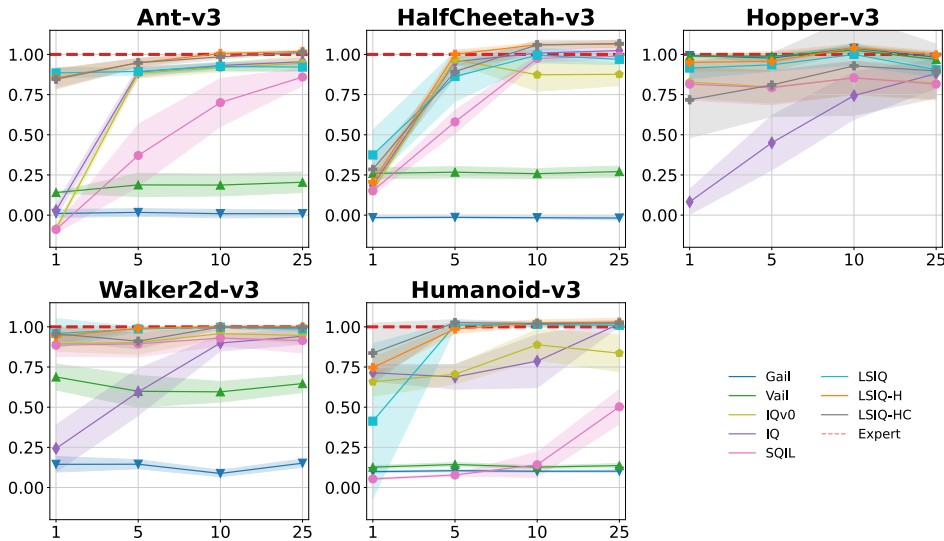

Figure 10: Comparison of the effect of the number of expert trajectories on different Mujoco environments. **States and actions** from the expert are provided to the agent. All plots are added here for the sake of completeness. Abscissa shows the normalized cumulative reward. Ordinate shows the number of expert trajectories. The first row shows the performance when considering states and action, while the second row considers the performance when using states only. Training results are averaged over five seeds per agent. The shaded area constitutes the 95% confidence interval. Expert cumulative rewards → Hopper:3299.81, Walker2d:5841.73, Ant:6399.04, HalfCheetah:12328.78, Humanoid:6233.45

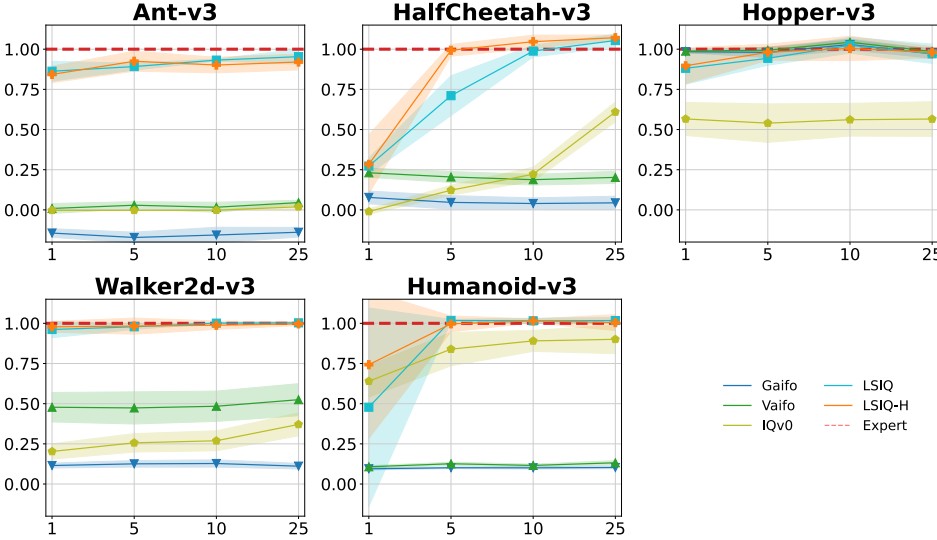

Figure 11: Comparison of the effect of the number of expert trajectories on different Mujoco environments. **Only expert states** are provided to the expert. All plots are added here for the sake of completeness. Abscissa shows the normalized cumulative reward. Ordinate shows the number of expert trajectories. The first row shows the performance when considering states and action, while the second row considers the performance when using states only. Training results are averaged over five seeds per agent. The shaded area constitutes the 95% confidence interval. Expert cumulative rewards → Hopper:3299.81, Walker2d:5841.73, Ant:6399.04, HalfCheetah:12328.78, Humanoid:6233.45

## C.5 IMITATION LEARNING FROM STATES ONLY – FULL TRAINING CURVES

The learning curves for the learning from observation experiments can be found in Figure 12, 13, 14 and 15 for 1, 5, 10 and 25 expert demonstrations, respectively.

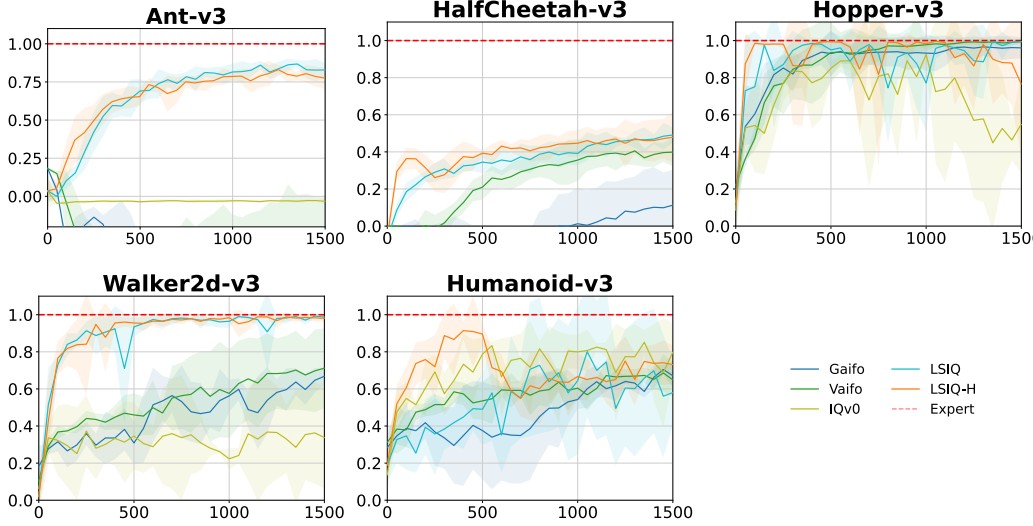

Figure 12: Training performance of different agents on Mujoco Tasks when using **1** expert trajectory consisting of **only states**. Abscissa shows the normalized discounted cumulative reward. Ordinate shows the number of training steps ($\times 10^3$). Training results are averaged over five seeds per agent. The shaded area constitutes the 95% confidence interval. Expert cumulative rewards → Hopper:3299.81, Walker2d:5841.73, Ant:6399.04, HalfCheetah:12328.78, Humanoid:6233.45

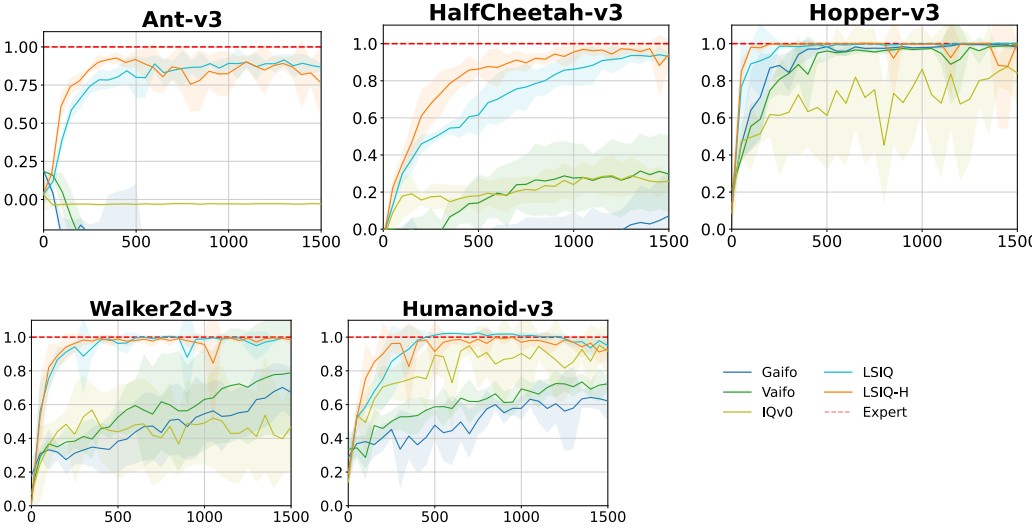

Figure 13: Training performance of different agents on Mujoco Tasks when using **5** expert trajectory consisting of **only states**. Abscissa shows the normalized discounted cumulative reward. Ordinate shows the number of training steps ($\times 10^3$). Training results are averaged over ten seeds per agent. The shaded area constitutes the 95% confidence interval. Expert cumulative rewards → Hopper:3299.81, Walker2d:5841.73, Ant:6399.04, HalfCheetah:12328.78, Humanoid:6233.45

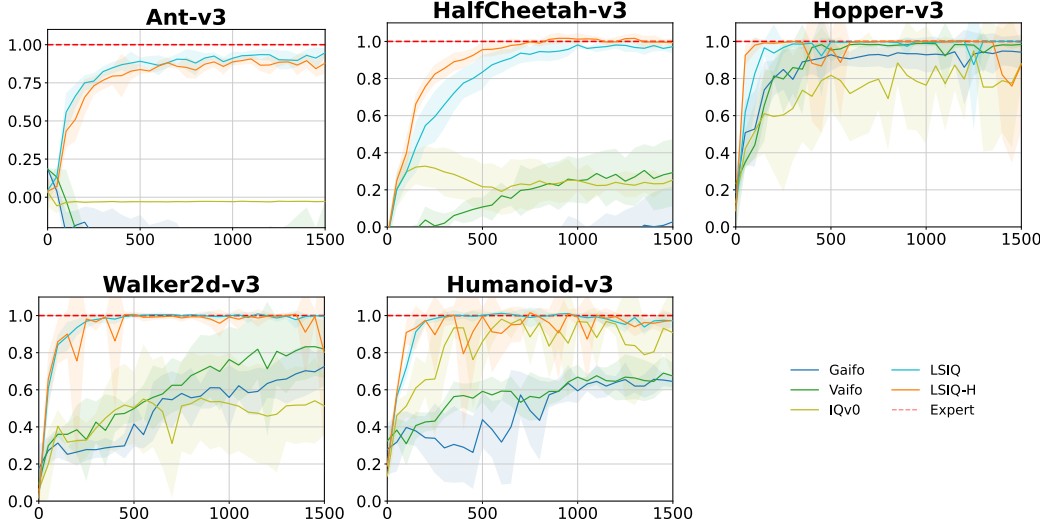

Figure 14: Training performance of different agents on Mujoco Tasks when using **10** expert trajectory consisting of **only states**. Abscissa shows the normalized discounted cumulative reward. Ordinate shows the number of training steps ($\times 10^3$). Training results are averaged over five seeds per agent. The shaded area constitutes the 95% confidence interval. Expert cumulative rewards → Hopper:3299.81, Walker2d:5841.73, Ant:6399.04, HalfCheetah:12328.78, Humanoid:6233.45

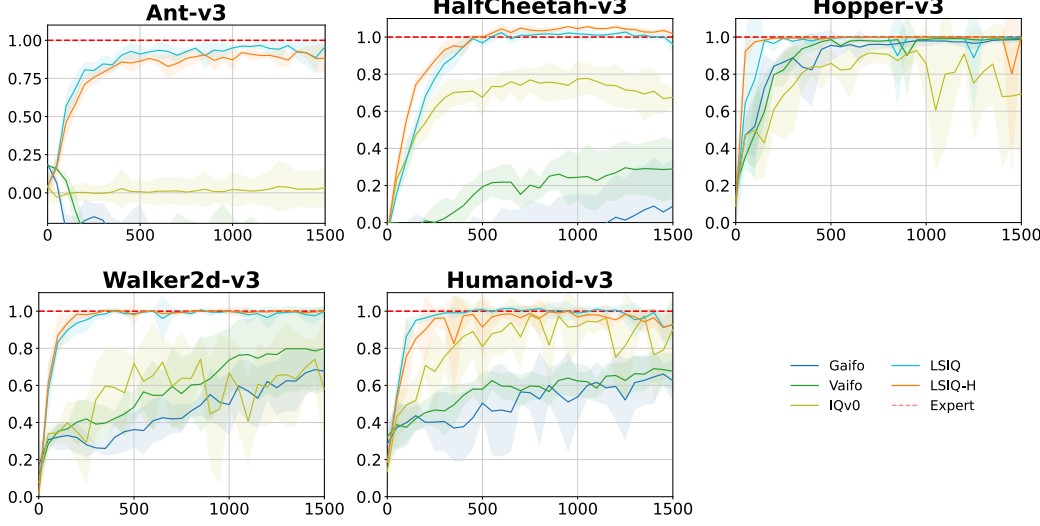

Figure 15: Training performance of different agents on Mujoco Tasks when using **25** expert trajectory consisting of **only states**. Abscissa shows the normalized discounted cumulative reward. Ordinate shows the number of training steps ($\times 10^3$). Training results are averaged over five seeds per agent. The shaded area constitutes the 95% confidence interval. Expert cumulative rewards → Hopper:3299.81, Walker2d:5841.73, Ant:6399.04, HalfCheetah:12328.78, Humanoid:6233.45

## C.6    IMITATION LEARNING FROM STATES AND ACTIONS – FULL TRAINING CURVES

The learning curves for the experiments where states and actions are observed can be found in Figure 16, 17, 18 and 19 for 1, 5, 10 and 25 expert demonstrations, respectively.

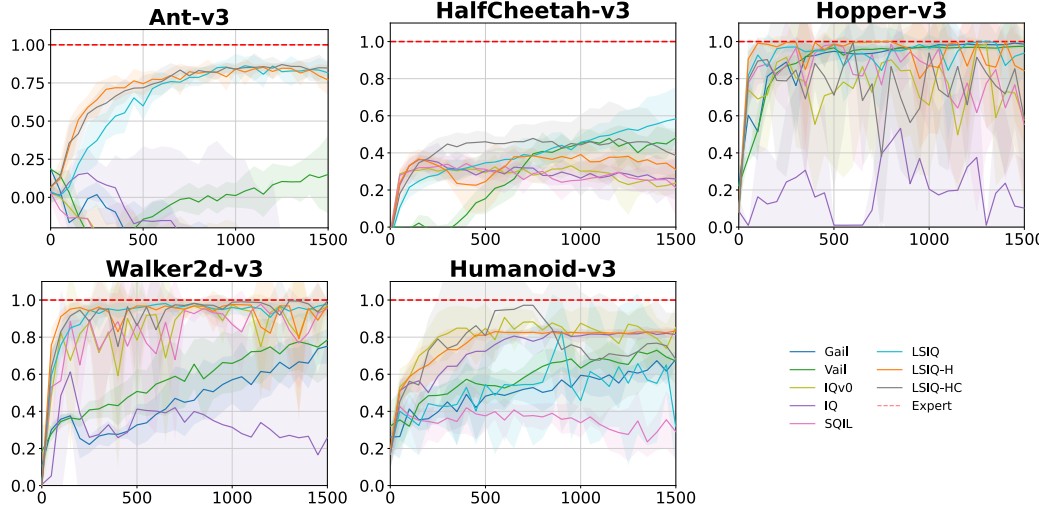

Figure 16: Training performance of different agents on Mujoco Tasks when using **1** expert trajectory. Abscissa shows the normalized discounted cumulative reward. Ordinate shows the number of training steps ($\times 10^3$). Training results are averaged over five seeds per agent. The shaded area constitutes the 95% confidence interval. Expert cumulative rewards → Hopper:3299.81, Walker2d:5841.73, Ant:6399.04, HalfCheetah:12328.78, Humanoid:6233.45

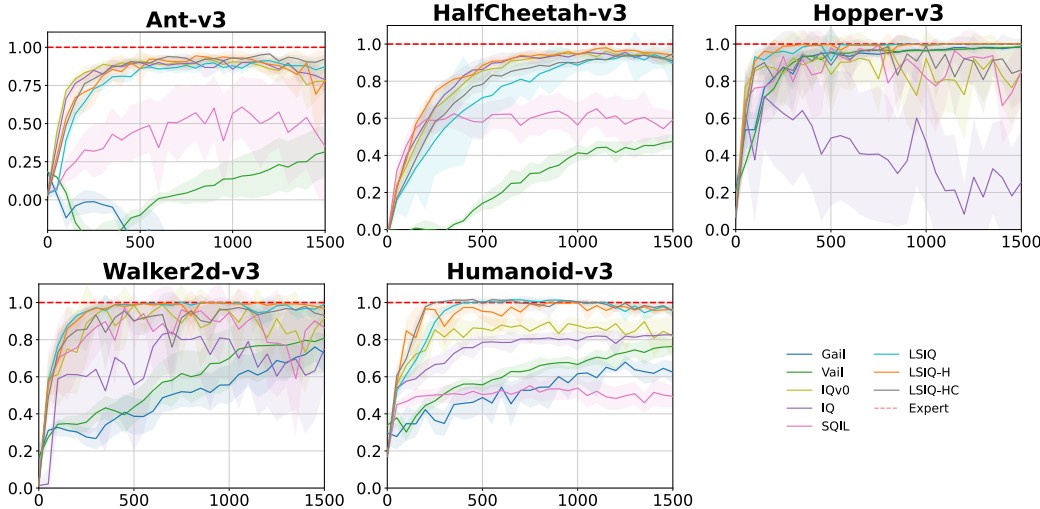

Figure 17: Training performance of different agents on Mujoco Tasks when using **5** expert trajectories. Abscissa shows the normalized discounted cumulative reward. Ordinate shows the number of training steps ($\times 10^3$). Training results are averaged over ten seeds per agent. The shaded area constitutes the 95% confidence interval. Expert cumulative rewards → Hopper:3299.81, Walker2d:5841.73, Ant:6399.04, HalfCheetah:12328.78, Humanoid:6233.45

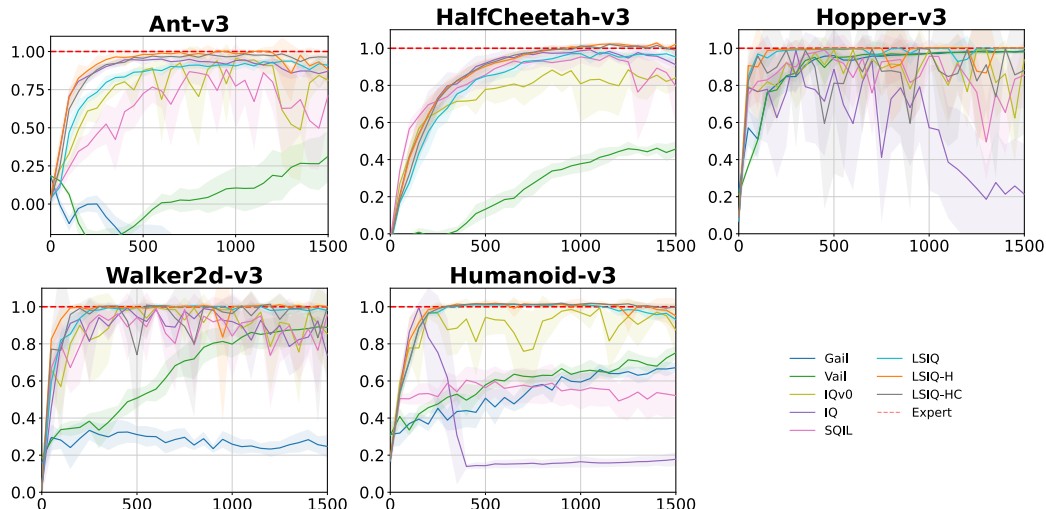

Figure 18: Training performance of different agents on Mujoco Tasks when using **10** expert trajectories. Abscissa shows the normalized discounted cumulative reward. Ordinate shows the number of training steps ($\times 10^3$). Training results are averaged over five seeds per agent. The shaded area constitutes the 95% confidence interval. Expert cumulative rewards $\rightarrow$ Hopper:3299.81, Walker2d:5841.73, Ant:6399.04, HalfCheetah:12328.78, Humanoid:6233.45

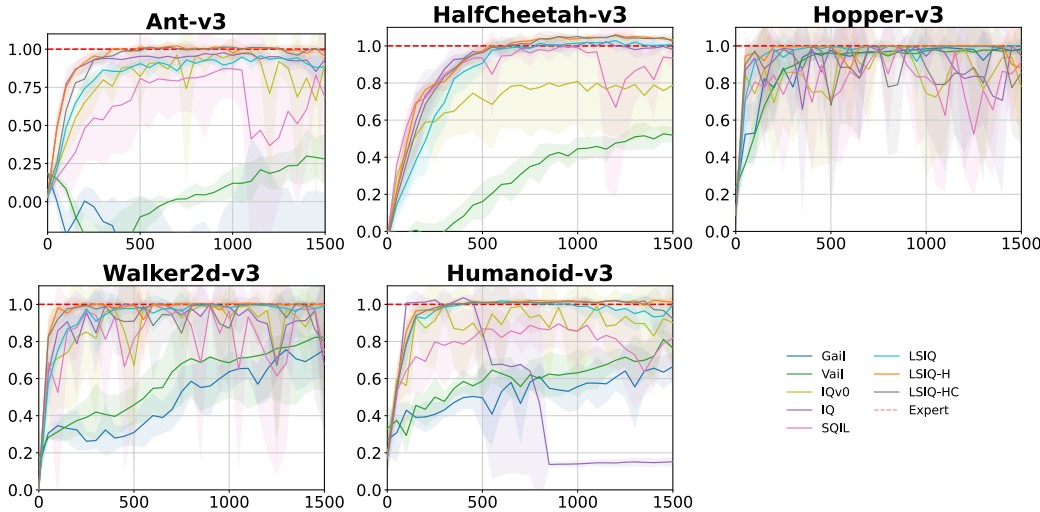

Figure 19: Training performance of different agents on Mujoco Tasks when using **25** expert trajectories. Abscissa shows the normalized discounted cumulative reward. Ordinate shows the number of training steps ($\times 10^3$). Training results are averaged over five seeds per agent. The shaded area constitutes the 95% confidence interval. Expert cumulative rewards $\rightarrow$ Hopper:3299.81, Walker2d:5841.73, Ant:6399.04, HalfCheetah:12328.78, Humanoid:6233.45

