# OpenReview forum: "LS-IQ: Implicit Reward Regularization for Inverse Reinforcement Learning"
_ICLR.cc/2023/Conference — ICLR 2023 poster_

### Official Review · Reviewer_BFcT · 2022-10-24

**Confidence:** 3
**Correctness:** 3
**Technical Novelty And Significance:** 2
**Empirical Novelty And Significance:** 2
**Recommendation:** 6

**Clarity, Quality, Novelty And Reproducibility:**

## Clarity

Most part of the paper are well written, but the clarity can be improved. Some parts are not clear. Give some suggesions bellow:
+ In Section 3.6, why not introducing your method using Fig. 2. How can a IDM training online with the collected data can infer action for expert observations? how to guarantee the correctness of the predicted expert actions?


## Quality
Motivated by theoretical analysis and derive the novel objective of LS-IQ, I can believe the quality of the proposed method is good.

## Novelty
The proposed method is novel.

## Reproducibility
Need more details to reproduce the results.


**Strength And Weaknesses:**

## Strength

+ This paper noticed that the squared norm regularization on the implicit reward function is effective in imitation learning, but lack of theoretical analysis. Interestingly, this paper uses the regularizer under a mixture distribution of the policy and the expert, and understand the learning procedure in a illuminating perspective. The original objective was understood as a least-squared Bellman error minimization, resulting an minimization of $\chi^2$-divergence between the expert and the mixture distribution.

+ Given theoretical analysis of the regularizer, this paper indicates some sources of instabilities of the IQ-Learn approach: the arbitrariness of the Q-function scale, exploding Q-function targets and reward bias. Then, Least Square Inverse Q-Learning was proposed to addressed these issues.


## Weakness

+ The proposed method are only evaluated on online imitation learning tasks. To fairly and fullly compare the proposed method to IQ-Learn, I suggest the authors to also evaluate the methods on offline settings as in the IQ-Learn paper.

+ The proposed method, LS-IQ, stabilize training by providing fixed targets for the Q-function on expert states, and properly treat absorbing states. This work also introduces an entropy-regulariation critic. However, there is no ablation to study the true effects of each component.


**Summary Of The Paper:**

This paper focus on addressing the reward bias issue of a SOTA imitation learning method called IQ-Learn that directly uses an implicit reward function. Specifically, this paper proposed an implicit reward regularization approach called Least Square Inverse Q-Learning (LS-IQ), which uses a modified inverse Bellman operator to address the reward bias issue of IQ-Learn. LS-IQ achieves this goal by using a squared L2 penalty on the mixture distribution of expert and policy. Empirically, LS-IQ outperforms some baseline methods on standard continuous control tasks.

**Summary Of The Review:**

Overall, the proposed method is well motivated by theoretical analysis. I expect the authors to further improve the clarity of the paper and address my concerns above.


======After rebuttal===
Thanks the authors for addressing my concerns. I agree to raise my rating to 6. Hope to see more details and open source code for better reproducibility.

---

> ### Author Response · Authors · 2022-11-18
> **Response to Reviewer BFcT**
>
> We thank the reviewer for pointing our attention to some crucial aspects of the empirical evaluation which we were missing in the first version of our paper. We focus most effort in recent days on enhancing the ablations and extending the experiments. We will provide the updated results in the experiment section and the appendix. We will also provide the code for all experiments in the supplementary material in the next hours.
>
> > The proposed method are only evaluated on online imitation learning tasks. To fairly and fully compare the proposed method to IQ-Learn, I suggest the authors to also evaluate the methods on offline settings as in the IQ-Learn paper.
>
> We want to point out that the original IQ-Learn Paper does not present results in the offline setting for the continuous action scenario, which is our setting. We extensively tested the original IQ-learn implementation in the offline scenario, without achieving any learning performance due to exploding Q-functions. While our paper tackles the issue of exploding Q-functions even in the offline setting, it does not tackle its specific challenges. Note that our regularizer explicitly depends on online data produced by the policy for regularization, which is why we switched to the original IQ regulaizer for the offline experiments. Extending our approach to the offline setting requires an in-depth analysis looking into this problem specifically. An interesting insight into this problem is presented by Li, Ziniu, et al. [1], where they highlight the importance of orthogonal regularization for the related ValueDice algorithm.
>
> As a starting point for future investigations, we will provide the code for the offline scenario for IQ and LS-IQ in the next hours.
>
> [1] *Li, Ziniu, et al. "Rethinking ValueDice: Does It Really Improve Performance?." arXiv preprint arXiv:2202.02468 (2022).*
>
>
> > The proposed method, LS-IQ, stabilize training by providing fixed targets for the Q-function on expert states, and properly treat absorbing states. This work also introduces an entropy-regulariation critic. However, there is no ablation to study the true effects of each component.
>
> We thank the reviewer for the suggestion. We performed an extensive ablation study and we will update the appendix with the results and the discussion. We hope that our ablation study clarifies the doubts of the reviewer. We will perform more ablations for the final version of the paper if the reviewer believes that some important comparison is missing.
>
> > In Section 3.6, why not introducing your method using Fig. 2.
>
> We appreciate the reviewer's suggestion, but unfortunately, we selected a more synthetic discussion due to space reasons.
>
> > How can a IDM training online with the collected data can infer action for expert observations? how to guarantee the correctness of the predicted expert actions?
>
> In the general scenario, an IDM cannot learn the exact expert action, as, in principle, multiple actions can lead to the same next state. However, this is not a major issue in MDPs as the current state contains all the information needed.
> Assuming that the effect of each action is distinct, that the capacity of the selected IDM is appropriate, that the policy data has some overlap with the expert one, and that we collect enough data, we can ensure that the predicted expert action is approximated with sufficient accuracy. Under these assumptions, the problem is equivalent to a supervised learning problem. Our empirical results support this claim.
>
> While most of these assumptions are common to every machine learning setting, the state distribution overlap is quite specific to this setting.
> While this assumption cannot be guaranteed in general for an arbitrary environment and an arbitrary initial policy, we argue that, in practice, the assumption holds  (at least locally) for two reasons: first, the initial state distribution is the same for the expert and policy. Second, the algorithm itself is pushing the policy towards expert state distribution.
>
> As the reviewer argues, a good extrapolation of expert action is fundamental to the performance of the algorithm. However, our empirical results prove that simple IDM models can extrapolate expert action with sufficient accuracy, at least in the proposed benchmarks.
>
> > Need more details to reproduce the results.
>
> We will add the code for the experiments in the supplementary material to resolve any reproducibility concerns.

---

### Official Review · Reviewer_XCLn · 2022-10-25

**Confidence:** 4
**Clarity, Quality, Novelty And Reproducibility:** 1. The original regularizer $\psi_{\p…
**Correctness:** 3
**Technical Novelty And Significance:** 4
**Empirical Novelty And Significance:** 3
**Recommendation:** 8

**Strength And Weaknesses:**

Strength
1. The paper is well-written, and I found the main contributions of this study to be novel.
2. The proposed method provides the correct way to deal with absorbing states in inverse reinforcement learning.
3. Show the relation between the proposed method, IQ-Learn, and SQIL.

Weakness
1. The proposed method is evaluated on the MuJoCo benchmarks, but they are relatively easy for modern IRL methods. It is unclear whether the proposed method is robust to distribution shifts between the expert and agent distributions.
2. As described below, some technical details are not shown in the manuscript. In particular, $r_{\min}(s)$ in Equation (20) should be explained.


**Summary Of The Paper:**

This paper provides a theoretical analysis of Inverse soft Q-Learn (IQ-Learn) and proposes a novel algorithm named Least Squares Inverse Q-learning (LS-IQ), which outperforms state-of-the-art algorithms such as GAIL, VAIL, IQ, and SQIL. At first, the authors show that the maximum entropy IR objective with the regularizer proposed by Garg et al. (2021) is identical to a minimization of a $\chi^2$ divergence between the expert state-action distribution and a mixture of expert and learner. Then, the authors formulate the IRL objective using the regularizer as the entropy-regularized least-squares problem, which is related to LSGAN. To derive the practical algorithm, the authors modify the loss function for the Q-function to incorporate the term added by the regularizer.

**Summary Of The Review:**

I think this paper studies an important problem and has some interesting theoretical results. Moreover, this paper is well-written, and the contribution is quite impressive.

---

> ### Author Response · Authors · 2022-11-18
> **Response to Reviewer XCLn**
>
> We thank the reviewer for the insightful comments. We tried to resolve most of the concerns of the reviewer and truly believe that, among others, the suggested additional experiments on more complex tasks enhance the quality of our paper. We appreciate the time and effort put in by the reviewer.
>
> > The proposed method is evaluated on the MuJoCo benchmarks, but they are relatively easy for modern IRL methods. It is unclear whether the proposed method is robust to distribution shifts between the expert and agent distributions
>
> We agree that the Mujoco Benchmarks have fine-tuned dynamics for RL agents, simplifying training. To prove that our algorithm can handle more challenging scenarios, we tested our approach on a complex locomotion task, using a simulated Atlas robot. In this scenario, the techniques presented in this paper provide a clear performance improvement w.r.t. other baselines. We will upload the related plots in the next hours.
>
> > As described below, some technical details are not shown in the manuscript. In particular, $r_\text{min}(s)$ in Equation (20) should be explained.
>
> This specific issue is resolved in the updated version of the paper. We hope that the improved notation resolves these concerns.
>
> > The original regularizer $\phi_{\pi\_{\text{E}}}(r)$ is recovered when $\alpha=1$. In this case, $r_{\text{min}}$ defined in Proposition 3.1 goes to negative infinity. Is it the main reason the original regularizer causes instabilities in continuous action spaces?
>
> It is correct that the true IQ-regularizer is recovered when $\alpha=1$ and that for $\alpha\rightarrow 1$ $r_{min} \rightarrow -\infty$, which is a direct consequence of the unbounded divergence and a clear source of instability in the continuous action scenario, where the policy is approximated with a neural-parameterized Gaussian.
>
> > The difference between Equations (13) and (14) is interesting. However, I am unsure whether $\tilde{Q}(s,a)$ converges or not when the agent policy converges to the expert policy. Although Figure 1 shows that the proposed method deals with absorbing states correctly, it would be better to discuss the convergence of $\tilde{Q}(s,a)$
> at the absorbing states.
>
> We thank the reviewer for the interesting and important question. We first want to point out, that the forward Bellman operator that underlies our inverse operator converges to the same fixed point as the standard Bellman operator. We agree that this important observation was missing in our initial submission. Therefore, we add the corresponding proof in Appendix A.5.
> Regarding the saddle-point convergence of the imitation learning algorithm, as given by Gnarg et al. (2021):
> We argue that these derivations should not be affected by our treatment of absorbing states because our inverse operator is consistent with the Bellman operator, also when not assuming absorbing state rewards of $0$.
>
> > In practice, sampling from $d_{\pi_\text{E}}$ or $d_{\pi}$ is infeasible. Would you explain how the training data, especially $\mathcal{D}_{\pi}$  is collected?
>
> As done in most reinforcement and inverse reinforcement learning approaches, we simply collect trajectory rollouts. Therefore, we are sampling from the (undiscounted) state distribution. We are aware that this is an approximation, but using the undiscounted state distribution allows us to sample and update the values of states that appear close to the end of the trajectory, avoiding the intractable number of rollouts needed to sample from $d_{\pi}$.
>
> > I do not fully understand the discussions given in Section 3.5. For example, if the minimization of Equation (13) is challenging, minimizing $E_{d_{\pi}} [(r_Q(s,a) - r_{\text{min}})^2]$ is also challenging. In addition, $r_{\text{min}}(s)$ is introduced in Equation (20), but it is not defined before. Is it a parameterized function of the state?
>
> The fixed target is a simple regression problem that does not rely on bootstrapping. Thus, the optimization landscape is simpler and makes it easier to learn the expert's action-value targets as suggested by the empirical results of our experiments. The mentioned issue about $r_{min}(s)$ is resolved in the updated version of the paper.

---

> > ### Comment · Reviewer_XCLn · 2022-12-13
> > **The manuscript has been much improved and is in a nice condition.**
> >
> > The authors have correctly addressed my comments. Additional experiments are nice, although I am unsure whether the proposed method can deal with noisy realistic data. The proof shown in Appendix A.5 is helpful for understanding the proposed algorithm.

---

### Official Review · Reviewer_kpRZ · 2022-10-25

**Confidence:** 4
**Correctness:** 2
**Technical Novelty And Significance:** 2
**Empirical Novelty And Significance:** 2
**Recommendation:** 5

**Clarity, Quality, Novelty And Reproducibility:**

For each category, I added my questions below:

### Clarity
- In introduction section, when implicit reward approach is introduced, Kostrikov's DAC is referred to be one of implicit reward approaches. Since DAC uses the discriminator, I think we should move it to explicit reward methods.
- In introduction, authors mentioned more robust extrapolation is possible, but I don't think any extrapolation was tested in the paper.
- In introduction, the "less variance" cannot be guaranteed through the experiments (I agree that the proposed method may stabilize the algorithm). Similarly, how can we see the exploding Q-functions targets?
- For IDM, there should be the references.
- In Introduction>Related Work> "aforementioned imitation learning approaches" need to be more clearly stated.
- In Section 2 in Eq (3), the definition of $\mathcal{J}_\rho$ is not defined.
- In Section 3.1., above Eq (7), it may be better to put a short definition of the variation form of $\chi^2$ divergence.
- In Section 3.1., below Eq (8), what is $r^*$?
- In Section 3.2., I think the authors argue that there are RL perspectives since we minimize Bellman errors, but it seems quite awkward to me. I think this is simply the form of the learning objective and not strongly relevant to RL.
- I couldn't understand the intuition in Section 3.3. The explanation between Eq (13) and Eq (14) needs to be more elaborated.
- In Section 3.3., Eq (13) and (14) uses subscripts iq and lsiq, but I would use non-italic capital letters.
- In Section 3.4., it is unclear why we have to introduce hard-Q for explanation. I think what authors have to emphasize is simply about using a single critic for entropy and reward regularizer, which is not relevant to hard-Q.
- Below Eq (17), I think the description should be elaborated.
- In the sentence "Finally, ~" below Eq (20). Why do we care about data imbalance and do we need to discuss this?


### Quality
- The partition function $Z_s$ below Eq (2) uses integral, but I think it should be defined by using summation since I believe the discrete action space is assumed in Notation section.
- In Eq (4), $r$ should be replace to the Bellman error in the last equation.
- In Eq (9), $\pi\in\Pi$ should be replaced to $\pi\in\Omega$.
- In the paragraph above (17), we need to remove "When now".


### Novelty
- Authors argue that the proposed objective is different from SQIL's objective in Eq (12). However, I believe if we use Q+c (constant) instead of Q in Eq (11), we can shift $r_{max}$ and $r_{min}$ by constant and find that the objective becomes exactly the same as that of SQIL.


### Reproducibility
- IQ-Learn is the crucial baseline of this work, and it seems to me that it performs far worse than what was reported from the IQ-Learn's paper. For example, when a single trajectory is used for Hopper in IQ-Learn paper, IQ-Learn successfully imitates the expert's performance. However, we can see that IQ-Learn performs poorly on Hopper in Figure 3 and 4. I wonder why such difference happened.

**Strength And Weaknesses:**

Strengths
- Authors look into the gap between practical algorithm and theoretical derivation of the baseline and try to minimize the gap.
- Algorithm is shown to applicable to various IL scenarios (online IL, LfO) and seems stable for all MuJoCo tasks.
- Algorithm used a lot existing techniques (dealing with absorbing states, IDM) to improve the performance.

Weaknesses
- The readability should be improved.
- Whether the proposed learning objective is new or not is unclear. I'd like to argue that the objective (12) has a strong relation with SQIL's objective.
- The baseline performance is much poorer than the performance reported in the paper.

**Summary Of The Paper:**

An online imitation learning algorithm with implicit reward is proposed, where the learning objective is to minimize $\chi^2$-divergence between the expert distribution and the mixture of expert and policy distributions. The idea of this work is motivated by the practical implementation of IQ-Learn that violates their theoretical derivation from introducing reward regularizer in practice. Authors argue that the proposed learning objective with the mixture-based divergence is more stable since it properly bounds the range of Q (whereas the range is  unbounded when the mixture is not considered). Additional details on absorbing states, a shared critic for entropy and reward regularization, and LfO with Inverse Dynamics Model are considered to improve the performance of the proposed idea.

**Summary Of The Review:**

Although the motivation of this work is interesting, the paper should become clearer to be accepted. With the current version, the authors' intuitions behind are not straightforward. There's also a baseline reproducibility issue.

---

> ### Author Response · Authors · 2022-11-18
> **Response to Reviewer kpRZ (Weaknesses)**
>
> We thank the reviewer for his very precise feedback and for correcting many details helping us to improve the overall quality of our work. We take particular care to address every point raised by the reviewer. We hope that this helps clarify our contribution, improving the readability of our work. In the following, we respond to every point raised by the reviewer. To address the concerns of the reviewer we will present further experiments and ablation studies and add them to the experiment section and the appendix in the next hours.
>
> > The readability should be improved.
>
> To improve the readability of our paper we restructured Section 3. We stressed the connection between the theoretical results and the proposed techniques. We hope that our claims are more clear and that the reading flow has improved.
>
> > Whether the proposed learning objective is new or not is unclear. I'd like to argue that the objective (12) has a strong relation with SQIL's objective. Authors argue that the proposed objective is different from SQIL's objective in Eq (12). However, I believe if we use Q+c (constant) instead of Q in Eq. (11), we can shift $r_{max}$ and $r_{min}$ by constant and find that the objective becomes exactly the same as that of SQIL.
>
> Indeed, the objective in (12) has a very strong connection with the one in SQIL. However, Equation (12) can express only symmetric rewards if $\alpha=0.5$ and there is no practical value for $\alpha$ that can recover exactly the SQIL objective, which is setting $r_\text{max}=1$ and $r_\text{min}=0$. As observed by the reviewer, the only difference between IQ and SQIL is a constant shift of the reward target.
> We want to point out that the LS-IQ objective is not the one presented in (12), but also considers the modifications described in the following sections of the paper. Indeed, by addressing the instabilities of the original IQ approach, our algorithm outperforms SQIL and IQ in the benchmarks. We want to remark that these increased performances are achieved thanks to the understanding of the connection between the two algorithms, guiding the design of each component. We added the full objective of LS-IQ in the appendix to avoid any misunderstanding.
>
> > The baseline performance is much poorer than the performance reported in the paper.
>
> We did not experience a notable performance gap between the IQ paper and our results. We made sure that our implementation matches the author's implementation. It is important to note that in the IQ paper some environments are using the telescoping loss function formulation -- IQv0 -- and some are using the objective as defined in our paper. In contrast to the IQ paper, we tested both variants on all environments. In the IQ-Learn paper, the environments "Humanoid-v3", "Hopper-v3", and "Walker2d-v3" were using IQv0, and "Ant-v3" and "HalfCheetah-v3" were using the objective presented in our paper, which we refer to as IQ.  Finally, It is also important to note that the performance also depends on the performance of the expert. Our experts were trained with SAC until convergence, yielding higher performance than the experts from the IQ paper, which makes the imitation learning tasks more challenging. For easy reproducibility, in the next hours, we will provide the code with experiment setups in the supplementary materials.

---

> > ### Author Response · Authors · 2022-11-18
> > **Response to Reviewer kpRZ (Clarity)**
> >
> > Here we will address all the issues regarding the clarity raised by the reviewer.
> >
> > > In introduction section, when implicit reward approach is introduced, Kostrikov's DAC is referred to be one of implicit reward approaches. Since DAC uses the discriminator, I think we should move it to explicit reward methods.
> >
> > We actually wanted to refer to Kostrikov's newer work, ValueDice, which is an implicit method. We thank the reviewer for spotting the mistake.
> >
> > > In introduction, authors mentioned more robust extrapolation is possible, but I don't think any extrapolation was tested in the paper.
> >
> > We thank the reviewer for the comment, we fixed the claim in the updated version of the paper.
> >
> > > In introduction, the "less variance" cannot be guaranteed through the experiments (I agree that the proposed method may stabilize the algorithm). Similarly, how can we see the exploding Q-functions targets?
> >
> > We present additional experiments in the appendix clarifying the problem of exploding $Q$-functions causing higher variance of the results in IQ.
> >
> > > For IDM, there should be the references.
> >
> > We are happy to include any further reference the reviewer suggests on IDMs. We reference most of the works on IDMs in the related section ("3.6 LEARNING FROM OBSERVATIONS") as they were helpful for the discussion and to keep the focus. We focus the related work section on the more central aspects of the paper.
> >
> > > In Section 3.2., I think the authors argue that there are RL perspectives since we minimize Bellman errors, but it seems quite awkward to me. I think this is simply the form of the learning objective and not strongly relevant to RL.
> >
> > In Section 3.2 we prove a strong relation between the SQIL and IQ algorithms. SQIL can be mostly seen as a direct reinforcement learning problem with fixed rewards for expert and policy transitions. While the resulting approach is not trivially a reinforcement learning method, its implementation is mostly rooted in the standard reinforcement learning approach and theory. We hope that this point is more clear in the updated version of the paper.
> >
> > > I couldn't understand the intuition in Section 3.3. The explanation between Eq (13) and Eq (14) needs to be more elaborated.
> >
> > We updated the respective section in the paper to make our explanation more intuitive and clear. We also added propositions to the appendix to clarify the soundness of the approach.
> >
> > > In Section 3.4., it is unclear why we have to introduce hard-Q for explanation. [.
> >
> > We want to point out that the introduction of the entropy and regularization critics is twofold: first, we need to switch to hard Q-functions to apply our proposed techniques while keeping the approach sound (this is not possible on soft Q-functions). Second, we introduce the regularization critic to perform the correct optimization problem for the policy, which was overlooked by the original IQ implementation.  We hope that the updated version of the paper is more clear about these points.
> >
> > > In the sentence "Finally, ~" below Eq (20). Why do we care about data imbalance and do we need to discuss this?
> >
> > We agree that data imbalance is not a major issue. Hence, we followed the reviewer's suggestion and removed the mentioned sentence.
> >
> >
> > We also addressed the minor comments of the reviewer (non-italic text, more explicit definition of $J_\rho$, etc.) in the updated version of the paper. We thank the reviewer again for this feedback that helped us further improve our manuscript.

---

### Official Review · Reviewer_GvAu · 2022-10-26

**Confidence:** 3
**Clarity, Quality, Novelty And Reproducibility:** Clarity is subpar. Quality, novelty, …
**Correctness:** 3
**Technical Novelty And Significance:** 3
**Empirical Novelty And Significance:** 3
**Recommendation:** 8

**Strength And Weaknesses:**

Strengths
- the biggest strength of the paper is that it is very practical: it identifies many tips and tricks for making IQ-learn better. This is a great contribution, given the already strong results of the base IQ-learn algorithm.
- some of these tricks are somewhat theoretically grounded. Though (as the paper notes itself) some parts are not super convincing, e.g. fixed Q target, it is still useful for future work.
- the experiments have ablations of using different subsets of the proposed tricks.

Weaknesses
The first few pages were find to read, but the rest of the paper (esp section 3) was poorly structured in my opinion. It reads like a laundry list A, B, C, and the reader has no idea what to anticipate next. Some of the insights feel like offhand remarks that have little relevance to the rest of the paper, and are hard to distinguish from the actual important insights. I would try to restructure the paper so that the most important parts are emphasized (even repeated), and move more minor details to the appendix.

**Summary Of The Paper:**

The paper studies the problem of imitation learning, building on the recent IQ-learn framework. Instead of an adversarial reward-policy loss like GAIL, IQ-learn instead parameterizes the Q-function so that the policy can be directly extracted. While IQ-learn works fine, the paper notes that some of the practical tricks don't match the analysis, e.g. regularizing both the expert and the imitator (which also should prevent direct extraction of policy in theory).

In this paper, the authors present Least Squares Inverse Q Learning (LS-IQ). LS-IQ patches some of the gap described above with IQ-learn.

- First, LS-IQ shows that the mixture regularizer is, naturally, regularizing a chi-squared divergence between the expert and the mixture of expert/policy occupancies. In practice this is good because the mixture ensures that the divergence is bounded.
- Second, they use a least-squares RL perspective to figure out how to properly treat absorbing states.
- Third, they propose using a regularization critic, to account for the additional regularizer term in the objective.
- A few more tips and tricks, e.g., replacing bootstrapping target with fixed target, learning from observations (with no action information) by training an IDM.


**Summary Of The Review:**

In summary the authors improve on IQ-learn, the current SOTA imitation learning algorithm on many tasks.
The paper presents a laundry list of tips and tricks, which was hard to read, but shows good practical improvements that should be very valuable to the community.

--------

Update: I have read the author response, and am keeping my evaluation.

---

> ### Author Response · Authors · 2022-11-18
> **Response to Reviewer GvAu**
>
> We thank the reviewer for the valuable suggestions and we are happy that the reviewer appreciates the practical contribution of our work.
>
> Following the reviewer's suggestions, we updated the paper to clarify the underlying connection between each technique. We restructured Section 3, where we describe the details of our approach. We highlighted the key ideas of our approach better and strengthen the connection between each contribution. In doing so, we stress more on the take-home message of our paper: the connection drawn between IQ and SQIL helped us understand the influence of the regularizer, and address the instability issues of IQ using techniques and assumptions deeply rooted in the classical reinforcement learning setting.
>
> As the reviewer suggested, we simplified the presentation of our method and moved details into the appendix. We believe that the reviewer's suggestions helped us increase the readability of the paper and improve the overall quality of the work. We hope that the reviewer is satisfied with the improvement and we are open to address further comments in future versions of the paper.
>
> We will also update the experiment section adding more environments and further ablations studies in the next hours.

---

### Decision · Program_Chairs · 2023-01-20

**Decision:**

Accept: poster

**Justification For Why Not Higher Score:**

There remain some clarification concerns among all reviewers.

**Justification For Why Not Lower Score:**

The problem identified is interesting, and the result is convincing. We believe the results worth publishing.

**Metareview: Summary, Strengths And Weaknesses:**

This paper identifies a few issues of a SOTA imitation learning---IQ-learn (such as reward bias), and propose a new algorithms LS-IQ which addresses these issues. Empirically, LS-IQ outperforms strong baseline methods on standard continuous control tasks.

All reviewers agree on that the problem studied is interesting, and the proposed algorithm is practical and achieve great performance, thus we are convinced with the contribution of this paper. The author response and the additional experiments added during the discussion phase also well addressed a majority of the reviewers concerns. We thus recommend acceptance.

**Note From Pc:**

if the above contains the word "oral" or "spotlight" please see: "oral" presentation means -> notable-top-5% and "spotlight" means -> notable-top-25%. As stated in our emails, we are disassociating presentation type from AC recommendations